# Leveraging Uncertainty of Pre-trained Models for Fine-Tuning with Search Engine Retrieval

## Abstract

Large pre-trained models can dramatically reduce the amount of task-specific data required to solve a problem, but they often fail to capture domain-specific nuances out of the box. The Web likely contains the information necessary to excel on any specific application, but identifying the right data a priori is challenging without knowing where the model is lacking in knowledge. This paper shows how to leverage recent advances in multi-modal learning to augment a pre-trained model with search engine retrieval. We propose to retrieve useful data from the Web based on instances the model is uncertain about. These uncertain cases are used without access to their labels to generate search queries with varying granularity of descriptiveness. For the final step of retrieval, we propose a geometry-aware refinement technique to discard images unrelated to the task. We demonstrate substantial performance improvements, e.g. a remarkable increase of 15 percentage points in accuracy on the Stanford-Cars and Flowers datasets while requiring two orders of magnitude less data compared to the state-of-the-art. We also present extensive experiments giving insights about what to expect of the proposed approach beforehand while exploring the impact of noisy retrieval and different learning strategies.

## 1 Introduction

The World Wide Web is the source of most training data for today's largest models. The abundance and diversity of the data available online has played a pivotal role in developing state-of-the-art models for vision, natural language processing (NLP), and multi-modal tasks (e.g., CLIP (Radford et al., 2021), GPT (Brown et al., 2020) LLama (Touvron et al., 2023)), ). Large pre-trained models have shown remarkable versatility across domains, and settings but their breadth often hides their limitations. Data quality and curation remain important (Fang et al., 2022; Nguyen et al., 2022), and increasing dataset sizes only masks the models' failure to generalize compositionally (Dziri et al., 2023). Even as models grow increasingly large, they inevitably contain knowledge gaps due to their static training data, creating an ongoing need for accessing up-to-date information.

One popular method for addressing some of the limitations of large pre-trained models is retrieval-augmented inference (Teney & Hengel, 2019; Guu et al., 2020a; Shuster et al., 2022). These methods retrieve instances at test time to produce the output. Existing methods rely on a supplementary (e.g., up-to-date or domain-specific) dataset, from which the model can retrieve information to supplement its pre-trained knowledge. Identifying relevant instances is challenging, these methods incur an increased computational cost at test time, and the limitations related to finite datasets remain (Wang et al., 2020; Xiong et al., 2021).

In contrast to the fixed static datasets of machine learning, humans continuously update their knowledge using new information, often acquired by querying the Web through search engines. The idea of connecting learning algorithms with the Web and search engines is not new but is challenging to implement effectively (Carlson et al., 2010a; Chen et al., 2013; Chen & Gupta, 2015; Shen et al., 2018; Teney & Hengel, 2019). We present a novel approach that exploits recent advances in NLP and multimodal learning (Jain et al., 2022; Girdhar et al., 2023; Dong et al., 2023) to significantly improve on the state-of-the-art.

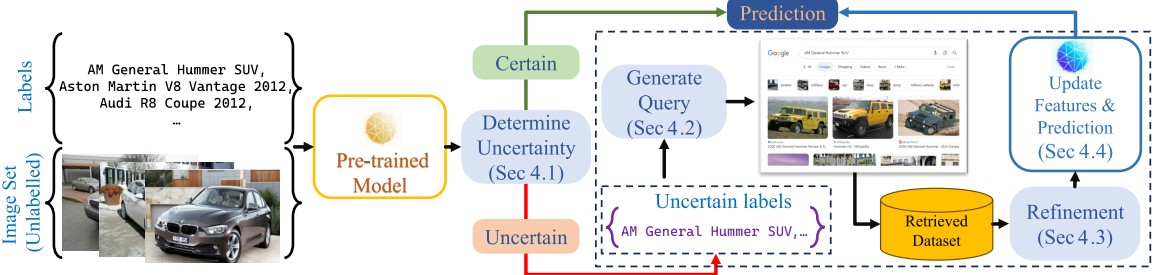

Figure 1: We propose a method to enrich a pre-trained visual recognition model using web search. We start by assessing the model's uncertainty about the given input and candidate labels, then issue a text query to acquire data for model updates. The predictions of the original and updated models are compared to assess information gained and generate the final output.

This paper examines how pre-trained models, which capture rich concept knowledge, can be enhanced using search engines that provide diverse information based on text queries. We introduce a method to identify uncertain cases before fine-tuning by generating search queries that describe the object's predicted attributes and scene context. These queries retrieve relevant images, which are then refined using a geometry-aware approach to filter out irrelevant data, as depicted in Figure 1. The refined images are used for minimal fine-tuning of the pre-trained model, improving its predictions with a compact task-specific model.

Contemporary to our efforts, Li et al. (2023) recently proposed Internet Explorer, a method for representation learning that also involves an active exploration of the Web. Their approach, formulated as reinforcement learning, samples concepts from ConceptNet (Speer et al., 2017) to uncover potentially relevant images. Our approach aims to be much more frugal and efficient in retrieving data. We use a more *focused* search strategy, and the process is only triggered when the pre-trained model shows high uncertainty while lowering the number of required images further using the proposed refinement technique.

On the other hand, our approach distinguishes itself from traditional active learning (Settles, 2010; Parvaneh et al., 2022) where the objective is to minimize the number of labels requested from annotators, given a fixed set of candidate examples. In contrast, our approach seeks new instances by formulating a textual search query. It then uses them as noisy, weakly-supervised examples to improve the model.

We observe that instances with low predictive entropy (i.e., deemed certain) remain largely unchanged after fine-tuning with retrieved images. This suggests that the pre-trained model may require adjustments in its conceptual decision boundaries to better handle nuanced instances and accurately classify the desired classes. We find that the model's representations allow us to clearly link the zero-shot task (the unlabeled images used for determining the uncertain cases and the list of all class labels for the task) and the improvement achieved from employing our approach. We further found if the task is to benefit from our approach, the best margin of improvement is achieved when the class name (provided by the dataset) is used as the search query rather than more complicated alternatives such as descriptions or instance-level queries from image captioning (Li et al., 2022b). This highlights the efficacy of contemporary search engines in aggregating related concepts and the generic class-level nature of datasets. Briefly, the contributions of this paper are:

- We propose a novel approach for enriching a pre-trained visual recognition model through access to web search with no need for additional labels or manual inputs. We show this approach integrates seamlessly into existing machine learning pipelines and improves performance.

- We implement the method on top of CLIP and derive a measure of uncertainty based on the classification entropy to make the method efficient in the amount of retrieved data. Technically, we use the projection of image embeddings onto a hypersphere and characterize the distribution of underlying concepts using a mixture of von Mises-Fisher distributions.

- We obtain significant improvements on the StanfordCars (Krause et al., 2013) and Flowers (Nilsback & Zisserman, 2008) datasets with improvements of over 15 percent in accuracy. We demonstrate a strong correlation between the observed improvements and the specific characteristics of the zero-shot task at hand. This correlation allows us to proactively assess whether the proposed search engine augmented

approach is advantageous for the task. If our approach proves unsuitable for the task, acquiring manually labeled data is a viable alternative.

## 2 Related Work

**Using unlabelled or weakly labelled data.** One area that shares some similarities with the proposed approach is active learning (Settles, 2010). In this paper, instead of relying on human experts, the proposed approach leverages the vast amount of knowledge available online. The proposed approach also connects to weakly supervised learning, where the aim is to learn from data that is only partially labelled (Zhou et al., 2018). In this case, the data collected from the search engine provides weak supervision for enhancing the model's performance. While weakly supervised learning often focuses on improving the label quality, the proposed approach aims to leverage the search engine as a source of inductive biases to improve the model's predictions.

**Cross-Modal retrieval.** Cross-modal retrieval is another area that is relevant to the proposed approach, where the goal is to retrieve relevant data from different modalities such as text, image, and video (Jiang et al., 2017). While cross-modal retrieval typically focuses on retrieving data from a fixed repository, the proposed approach enables real-time, up-to-date, and relevant information to be retrieved from the Internet using natural language queries.

**Language models.** The success of the proposed approach also relies on recent advances in language models, such as OpenAI's GPT-3, which enables machines to interact and process natural language text (Brown et al., 2020). Jointly learning image and text in models such as OpenAI's CLIP (Radford et al., 2021) or Google's ALIGN (Jia et al., 2021) have shown promise for improving the performance of visual models. In addition, self-supervised learning and generative modelling have shown promise for learning from noisy labels (Chen et al., 2020). By utilizing these technologies, the proposed approach takes a significant leap forward that otherwise would not be possible.

**Retrieval-augmented.** Retrieval-augmented learning offers an exciting avenue for enhancing the capabilities of models in an era where access to vast and dynamic information is paramount. For instance, Guu et al. (2020b;a); Shuster et al. (2022) learn to leverage large additional data sources for predictions. However, that requires using a common embedding space and implementation of indexing mechanisms. Xie et al. (2023) on the other hand, train independent retrieval networks alongside attention modules on top of CLIP while using large static datasets as sources of retrieval.

**Webly-supervised learning.** NELL (Carlson et al., 2010b) and NEIL (Chen et al., 2013) are amongst the first to explore acquiring new concepts and relationships that are periodically being refined with human supervision from the Web. Ideas from webly-supervised learning, where the focus is on harnessing the vast and noisy data available on the Web have lent ideas for large pre-training (Guo et al., 2018; 2017; Dai et al., 2018). In Beser & Bulling (2021) and Li et al. (2018) webly supervised zero-shot learning in natural language processing is explored. Further, Zhang et al. (2017) demonstrates how web data can enhance fine-grained categorization tasks. Zang et al. (2019); Zhang et al. (2017) combines webly supervised and zero-shot learning techniques for fine-grained recognition tasks. Together, these works offer valuable insights into leveraging web data for training machine learning models. However, using search engines where text is the medium for representation remains under-explored.

## 3 Multimodal Pre-trained Models

A large pre-trained model such as CLIP is trained using a self-supervised objective to align the representations obtained from both language and image outputs, respectively. The significance of this is that after training, we can map either textual or visual inputs using their corresponding neural encoders $\phi_t, \phi_x$ to the same semantic space. One way to interpret these models is to consider an image $\boldsymbol{x} \in \mathcal{X}$ and its corresponding textual description $\boldsymbol{t} \in \mathcal{T}$ as $p(\boldsymbol{x}, \boldsymbol{t}) \propto \exp(\langle \phi_x(\boldsymbol{x}), \phi_t(\boldsymbol{t}) \rangle)$, where $\langle \phi_x(\boldsymbol{x}), \phi_t(\boldsymbol{t}) \rangle$ represents the cosine similarity between visual and textual embeddings in the joint semantic space.

Here, $\mathcal{X}$ and $\mathcal{T}$ denote the space of image and text collected in a large corpus. Then, for the downstream classification task with the label set $\mathbf{Y}$, we have:

$$p_{\text{pre}}(\boldsymbol{y} \mid \boldsymbol{x}) = \frac{\exp\left(\alpha \cdot \langle \phi_x(\boldsymbol{x}), \phi_t(\boldsymbol{y}) \rangle\right)}{\sum_{\boldsymbol{y}' \in \mathbf{Y}} \exp\left(\alpha \cdot \langle \phi_x(\boldsymbol{x}), \phi_t(\boldsymbol{y}') \rangle\right)}. \tag{1}$$

Where $\alpha$ represents the inverse temperature controlling prediction sharpness and the labels $\boldsymbol{y}$ are used in their textual form, benefiting from the inherent capability of such pre-trained models for open-ended problems. The predicted label is therefore derived as $y^\star = \arg\max_{\boldsymbol{y}' \in \mathbf{Y}} p_{\text{pre}}(\boldsymbol{y}' \mid \boldsymbol{x})$.

## 4  Search Engine Augmented Learning

Our main goal is to improve the performance of a pre-trained model at inference time, i.e., CLIP. In our setting, we are given an (unlabelled) "target" dataset $\mathbf{X}$ of images and a list of potential labels $\mathbf{Y}$. We are interested in training a complementary model on a "small" retrieved dataset to predict the target labels. To build such a dataset, we leverage the vast amount of data on the Internet by creating a retrieval mechanism (note there is no curated labelled training set). We propose a retrieval mechanism that considers the uncertainty in the classification of the pre-trained model, queries and downloads images from search engines like Google. Our retrieval mechanism aims to effectively cover the uncertain classes in the target dataset. The retrieved dataset is expected to be noisy, containing unrelated images for certain class names. For example, one can look for images of flowers with the class name "Prince of Wales feathers" and retrieve images of royalty instead of flowers. In light of this, our retrieval mechanism also incorporates a refinement process to clean the dataset before training a classifier. We consider the following steps as the general algorithm:

1. determine the uncertain instances in the given unlabelled target dataset (see section 4.1)
2. formulate a query and invoke the search engine to retrieve the relevant images (see section 4.2)
3. refine and filter out the unrelated images (see section 4.3)
4. train a small model (e.g., a linear probe) for classification (see section 4.4)

Formally, for the set of uncertain labels $\mathbf{Y}_{\text{uncertain}}$, with access to a search engine (SE) that allows sampling $n$ most appropriate images given an uncertain class label $\boldsymbol{y}_r$, we consider:

$$\boldsymbol{\theta}^\star = \arg\max_{\boldsymbol{\theta}} \frac{1}{n} \sum_{(x_r, y_r) \in \mathcal{R}} \log\left(p_{\boldsymbol{\theta}}(\boldsymbol{y}_r \mid \boldsymbol{x}_r) \cdot \delta(x_r \mid \mathbf{X})\right), \tag{2}$$

$$\text{where } \mathcal{R} = \{(\boldsymbol{x}_r, \boldsymbol{y}_r) \mid \boldsymbol{x}_r \sim \text{SE}(\mathbf{y}_r), \forall \boldsymbol{y}_r \in \mathbf{Y}_{\text{uncertain}}\}. \tag{3}$$

Where $\boldsymbol{\theta}$ is a small number of parameters compared to the ones in the pre-trained model (e.g., a linear classifier optimized on a frozen CLIP backbone) and $\delta(\boldsymbol{x}_r, \mathbf{X})$ is a binary function indicating whether a given sample $\boldsymbol{x}_r$ belongs to the same distribution as those in $\mathbf{X}$ (see section 4.3). The optimal inference-time $\boldsymbol{\theta}^\star$ are the maximum likelihood parameters for the set $\{(\boldsymbol{x}_r, \boldsymbol{y}_r)\}$ obtained from the search engine. Effectively, a search engine enables us to *sample from the distribution* of all images potentially relevant to a query $\boldsymbol{y}_r$.

### 4.1  Uncertainty in pre-trained models

To construct the dataset, we focus on the image samples whose label is uncertain for the pre-trained model. Considering the conditional probability in equation 1, we can use the normalized Shannon entropy as a measure of uncertainty, with a threshold $\tau_H$ to construct a subset of uncertain instances:

$$\mathbf{X}_{\text{uncertain}} = \{\boldsymbol{x} \mid \mathbb{H}_{\text{pre}}(\boldsymbol{y} \mid \boldsymbol{x}) \geq \tau_H, \forall \boldsymbol{x} \in \mathbf{X}\}. \tag{4}$$

Where $\mathbb{H}$ represents the entropy value. Note that this uncertainty measures the uniformity of predictions for the given labels using the pre-trained model. If the model is not confident, i.e., produces a higher score for a class, this uncertainty will be high. We collect samples with entropy above a threshold, indicating the need for additional information for accurate prediction. We pick all the class labels in the top-$k$ predictions for the uncertain instances $\boldsymbol{x} \in \mathbf{X}_{\text{uncertain}}$ and create an uncertain label set $\mathbf{Y}_{\text{uncertain}}$. This uncertainty-based selection strategy is theoretically justified in Appendix A, where we prove that instances with higher predictive entropy provide greater expected information gain from retrieved data.

## 4.2 Creating a search engine based dataset

With the uncertain label set identified, we can now generate queries to obtain relevant images from the search engine. In this stage, we leverage the uncertain class labels to construct a query for the search engine. We subsequently collect the retrieved images. We consider the following strategies for search queries for each uncertain class:

1. **Classnames** ($\mathbf{D}^{\texttt{cls}}_{\text{uncertain}}$): Directly using "`{class_name}`" which is the name given to the class in the dataset (e.g., CLIP Benchmark (CLIP Benchmark, 2022)). In other words, for a specific uncertain label, we simply search for the class name.
2. **Descriptions** ($\mathbf{D}^{\texttt{desc}}_{\text{uncertain}}$): Employing an LLM, i.e., GPT-3, to generate "{description}"s for the class names similar to Menon & Vondrick (2023). For each class, we generate multiple descriptions. Then, the class names are concatenated with these descriptions, i.e., "`{class_name} which (is/has/etc) {description}`" to produce a search query.
3. **Captioning** ($\mathbf{D}^{\texttt{cap}}_{\text{uncertain}}$): Using a captioning methods such as BLIP (Li et al., 2022b). The class name is then concatenated with this caption as "`{class_name} {caption}`". Note that in this approach, the query is constructed for each instance individually as opposed to the previous two alternatives.

We can now create a dataset of retrieved images. For instance, for **Classnames**, we have:

$$\mathbf{D}^{\texttt{cls}}_{\text{uncertain}} = \Big\{ (\boldsymbol{x}_r, \boldsymbol{y}_r) \Big| \, \boldsymbol{x}_r \sim \text{SE}(\boldsymbol{y}_r), \forall \boldsymbol{y}_r \in \mathbf{Y}_{\text{uncertain}} \Big\}. \tag{5}$$

Here, $\boldsymbol{y}_r$ denotes the search query obtained using one of the strategies above (e.g. classnames). We use $\mathbf{D}_{\text{uncertain}}$ to denote any dataset constructed from the above-mentioned strategies.

## 4.3 Refinement

Ideally, the samples in $\mathbf{D}_{\text{uncertain}}$ should consist of images from a distribution similar to that of the target dataset, yet possessing distinctive features that can enhance performance. However, either due to query ambiguity or dataset specificity, there might be images in $\mathbf{D}_{\text{uncertain}}$ that do not conform to the distribution of those in $\mathbf{X}$. Therefore, a refinement step is required to discard the potentially noisy samples and create a refined subset $\mathbf{D}_{\text{refined}}$. To this end, we propose an approach to compare the distributions of the retrieved and target datasets.

Instead of adopting complex density estimation models, such as those necessitating the training of additional neural networks (e.g., , Kingma & Welling (2013); Abbasnejad et al. (2020; 2019); Ho et al. (2020)), we note CLIP's image embeddings, like many other contrastive learning alternatives, are inherently normalised to the unit hypersphere (Shi et al., 2023) (see Figure D.9). Thus, we leverage the mixture of von Mises-Fisher Distributions (MoVMF) (Banerjee et al., 2005) to model this hypersphere distribution. The mixture of von Mises-Fisher distributions is a statistical model that combines multiple von Mises-Fisher (VMF) distributions to represent data distributed on a hypersphere in high-dimensional space, allowing for the modelling of complex directional data patterns. Once optimised, MoVMF allows us to represent the target distribution as $p(\boldsymbol{x} \mid \mathbf{X}) = \sum_i \pi_i p_{\boldsymbol{\mu}_i, \boldsymbol{\kappa}_i}(\boldsymbol{x})$, where $\pi_i$ represents the mixing coefficient, and $\boldsymbol{\mu}_i$ and $\boldsymbol{\kappa}_i$ are the parameters of the distribution. We then use the most likely component to assign a hard cluster for each instance. Subsequently, we only keep samples that are sufficiently close to the mean of each vMF component:

$$\delta(\boldsymbol{x}_r \mid \mathbf{X}) = \mathbb{I}\big[\langle \phi_x(\boldsymbol{x}_r), \boldsymbol{\mu}_{i^*} \rangle < \tau_R \big], \tag{6}$$

$$i^* = \arg\max_i \, p_{\boldsymbol{\mu}_i, \boldsymbol{\kappa}_i}(\boldsymbol{x}_r). \tag{7}$$

During refinement, we found that using images, either from $\mathbf{X}_{test}$ or $\mathbf{X}_{train}$, without access to labels resulted in rejecting the same samples. We also explore two alternative approaches (see Appendix E), but empirical results demonstrate MoVMF performs better, is straightforward to implement, and imposes minimal assumptions on the data.

## 4.4 Prediction

In the final step, we train a linear classifier with parameters $\boldsymbol{\theta}$ on the refined retrieved samples. If during inference, there is no information gained (i.e., information gain or mutual information of label and additional

Table 1: Geometric analysis of the CLIP feature space.

|  | Pets | Flowers | Cars | Food | ImageNet |
|---|---|---|---|---|---|
| Label Span | 0.2885 | 0.2620 | 0.3298 | 0.4496 | 0.1579 |
| Dataset-$\mathbf{D}_{\text{uncertain}}$ Modality Sim. | 0.9618 | 0.9836 | 0.9889 | 0.8943 | 0.9671 |
| Dataset-$\mathbf{D}_{\text{uncertain}}$ Class Sim. | 0.9540 | 0.9590 | 0.9800 | 0.8930 | 0.9350 |

data is not positive), we fall back to the pre-trained model. In other words, we only use this new model if it leads to a reduced entropy (i.e., the confidence in predictions improves):

$$
\boldsymbol{y}^* = \arg\max_{\boldsymbol{y}} \begin{cases} p_{\boldsymbol{\theta}^\star}(\boldsymbol{y} \mid \boldsymbol{x}), & \text{if } \text{IG}(\boldsymbol{y} \mid \boldsymbol{x}, \mathbf{D}_{\text{refined}}) \geq 0 \\ p_{\text{pre}}(\boldsymbol{y} \mid \boldsymbol{x}), & \text{otherwise} \end{cases} . \tag{8}
$$

Where $\text{IG}(\boldsymbol{y} \mid \boldsymbol{x}, \mathbf{D}_{\text{refined}}) = \mathbb{H}_{\boldsymbol{\theta}^\star}(\boldsymbol{y} \mid \boldsymbol{x}, \mathbf{D}_{\text{refined}}) - \mathbb{H}_{\text{pre}}(\boldsymbol{y} \mid \boldsymbol{x})$ is the information gain, $\boldsymbol{y}^*$ is the predicted class for $\boldsymbol{x}$, and $\mathbb{H}_{\boldsymbol{\theta}^\star}(\boldsymbol{y} \mid \boldsymbol{x}, \mathbf{D}_{\text{refined}})$ denotes the entropy of the classifier (i.e., the entropy once the model is updated). The theoretical foundation for this information-theoretic approach is detailed in Appendix A, which establishes the optimality of uncertainty-based retrieval under budget constraints.

## 5 Experiments

We employ CLIP (Radford et al., 2019) as the pre-trained model to validate our approach while under the assumption of not having access to a labeled training dataset. Consequently, our initial comparison revolves around assessing the performance against the baseline zero-shot CLIP. Nevertheless, we also provide supplementary experiments exploring alternative zero or few-shot settings, which can be found in Appendix K for a more comprehensive understanding of our approach. We use NVIDIA A100 GPUs for all experiments.

### 5.1 Datasets and metrics

We evaluate our method on five popular image classification datasets: Flowers (Nilsback & Zisserman, 2008), Pets (Parkhi et al., 2012), Stanford Cars (Krause et al., 2013), Food (Bossard et al., 2014) and ImageNet (Deng et al., 2009). These datasets consist of 102, 37, 196, 101 and 1000 classes. We evaluate the impact of the retrieved dataset $\mathbf{D}_{\text{uncertain}}$ by training a linear probe and combining its predictions with those made by CLIP using the accuracy metric. We retrieve 100 images from the search engine for constructing $\mathbf{D}_{\text{uncertain}}^{\text{cls}}$. See Appendix F for technical details on retrieval. For each class, on average, 5 descriptors were generated using (Menon & Vondrick, 2023). For each descriptor, we retrieve 100 images to construct $\mathbf{D}_{\text{uncertain}}^{\text{des}}$. Due to the instance-level captioning approach for construction of $\mathbf{D}_{\text{uncertain}}^{\text{cap}}$, we retrieve only the top-5 images to target the closest images to the given caption-based query. For hyperparameters, we perform grid search using a validation set. Refinement threshold ranges from 0.1 to 0.7 and the learning rate is searched over on a logarithmic scale. The accuracy remains consistent across different seeds.

### 5.2 What tasks are expected to benefit?

Before using $\mathbf{D}_{\text{uncertain}}$ for classifier training, we evaluate the quality of retrieved images by measuring their distance from the dataset in CLIP's feature space which informs us of any domain shifts between the two sets of images. When datasets contain labels with higher specificity, the domain shift between the retrieved images and the dataset is expected to be reduced. So we use the span of the text modality as a proxy for specificity. These values are detailed in Table 1. The label span is derived using the cosine similarity between the two most distant label names in the CLIP feature space. Similarly, the dataset-to-image similarity is determined using the similarity between the mean vectors of each image set (titled modality similarity in Table 1). Another similar approach can be taken where instead of the entire image set, images of the same class from the retrieved dataset can be compared to the dataset images (titled class similarity in Table 1). In terms of the label span, ImageNet obtains the broadest span (lowest similarity), as expected given its large number of classes. In contrast, the span of labels in Food forms the narrowest cone in the CLIP feature space among all the studied datasets suggesting either low specificity or insufficient training data per category. As

Table 2: Results of training with retrieval techniques $\mathbf{D}^{\texttt{cls}}_{\text{uncertain}}$, $\mathbf{D}^{\texttt{cap}}_{\text{uncertain}}$, $\mathbf{D}^{\texttt{desc}}_{\text{uncertain}}$ and the impact of refinement alongside the retrieved number of images.

| Dataset | Number of Images (Unrefined / Refined) | | | Accuracy Results (%) (Unrefined / Refined) | | | | |
|---|---|---|---|---|---|---|---|---|
| | $\mathbf{D}^{\texttt{cls}}_{\text{uncertain}}$ | $\mathbf{D}^{\texttt{cap}}_{\text{uncertain}}$ | $\mathbf{D}^{\texttt{des}}_{\text{uncertain}}$ | ZS | $\mathbf{D}^{\texttt{cls}}_{\text{uncertain}}$ | $\mathbf{D}^{\texttt{cls+cap}}_{\text{uncertain}}$ | $\mathbf{D}^{\texttt{cls+des}}_{\text{uncertain}}$ | $\mathbf{D}^{\texttt{cls+cap+des}}_{\text{uncertain}}$ |
| Pets | 4096 4011 | 50653 50428 | 23226 22973 | 89.04 | 92.61 ↑3.57 92.64 ↑3.60 | 92.45 ↑3.41 92.56 ↑3.52 | 92.59 ↑3.55 92.67 ↑3.63 | 92.78 ↑3.74 **93.00** ↑3.96 |
| Flowers | 11197 10938 | 27932 27506 | 79508 75111 | 71.15 | 86.24 ↑15.09 86.32 ↑15.17 | 85.44 ↑14.29 86.27 ↑15.12 | 85.77 ↑14.62 86.09 ↑14.94 | 86.36 ↑15.21 **86.62** ↑15.47 |
| Cars | 21514 19416 | 149820 139458 | 156491 120293 | 64.71 | 80.52 ↑15.81 80.48 ↑15.77 | 80.33 ↑15.62 81.82 ↑17.11 | 80.81 ↑17.10 80.79 ↑17.08 | 82.32 ↑17.61 **82.38** ↑17.67 |
| Food | 11100 7062 | 619521 362849 | 56929 31228 | 88.73 | **88.97** ↑0.24 88.35 ↓-0.38 | 88.89 ↑0.16 88.87 ↑0.14 | 88.87 ↑0.14 88.63 ↓-0.10 | 88.86 ↑0.13 88.82 ↑0.09 |
| ImageNet | 198044 21725 | 558868 25000 | 502898 61160 | 68.33 | 69.05 ↑0.72 67.04 ↓-1.29 | 69.54 ↑1.21 69.78 ↑1.45 | 69.80 ↑1.47 69.73 ↑1.40 | 70.14 ↑1.81 **70.33** ↑2.00 |

Table 3: Comparison of training a LinearProbe (LP) using (1) the labelled training set ($\mathbf{T}_r$) and (2) the training set ($\mathbf{T}_r$) augmented with the retrieved dataset $\mathbf{D}_{\text{uncertain}}$.

| | Flowers | Pets | Cars | Food | ImageNet |
|---|---|---|---|---|---|
| *ZS ViT-B-16* | 71.15 | 89.04 | 64.71 | 88.73 | 68.33 |
| Ours | 86.62 ↑15.47 | 93.00 ↑3.96 | 82.38 ↑17.67 | 88.97 ↑0.24 | 70.33 ↑2.00 |
| LP w/ $\mathbf{T}_r$ | 96.41 | 92.72 | 85.64 | 92.08 | 78.66 |
| LP w/ $\mathbf{D}_{\text{uncertain}} + \mathbf{T}_r$ | 96.57 ↑0.16 | 93.62 ↑0.9 | 86.03 ↑0.39 | 92.15 ↑0.07 | 78.94 ↑0.28 |

for the image similarities in Table 1, the features of retrieved images for Cars and Flowers are the closest to those of the target dataset. Meanwhile, the noticeable deviation for the same value for Food indicates the retrieved images for this dataset might not be as informative for the classifier due to domain shifts between the dataset and retrieved images, leading to a smaller accuracy improvement.

### 5.3 On the different retrieved datasets

Table 2 shows the accuracy comparison of the proposed query constructions $\mathbf{D}^{\texttt{cls}}_{\text{uncertain}}$, $\mathbf{D}^{\texttt{cap}}_{\text{uncertain}}$ and $\mathbf{D}^{\texttt{desc}}_{\text{uncertain}}$, section 4.2, to model's ZeroShot accuracy (ZS) and the effect of the refinement process. For $\mathbf{D}^{\texttt{desc}}_{\text{uncertain}}$, we use the descriptions from Menon & Vondrick (2023).

The refinement process does not always improve the performance of the initially retrieved dataset. Excluding Food, this approach consistently works for datasets constructed with more diverse queries like $\mathbf{D}^{\texttt{cap}}_{\text{uncertain}}$. The images tend to be more diverse and cover more of the real distribution of each label when we construct datasets $\mathbf{D}^{\texttt{cap}}_{\text{uncertain}}$ and $\mathbf{D}^{\texttt{desc}}_{\text{uncertain}}$. However, those procedures also add more noisy instances. Using refinement, we can retrieve more samples and simultaneously remove ambiguous images that could decrease the performance of our method. Except for the retrieved datasets $\mathbf{D}^{\texttt{cls}}_{\text{uncertain}}$ with refinement for Food and ImageNet, we obtain improvements compared to zero-shot ViT-B-16 when we use the retrieved dataset. This is noticeable for the cases of Flowers and Cars, where our method obtains an accuracy improvement on average of 14.92% and 16.01%, respectively.

### 5.4 Comparison with SOTA

We compare our method with Internet Explorer (IE) (Li et al., 2023), SuS-X (Udandarao et al., 2023), and RECO (Iscen et al., 2023) which are SOTA methods directly comparable to our approach, as they do not modify the pre-trained model's structure. Due to the implementation of each SOTA on different CLIP

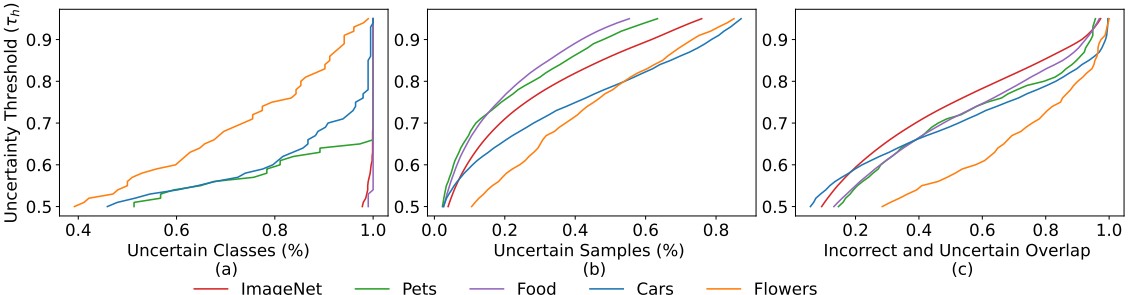

Figure 3: Uncertain plots for each dataset: (a) percentage of uncertain instances, (b) the ratio of uncertain classes, and (c) the overlap between incorrect and uncertain instances, against the uncertainty threshold $\tau_H$.

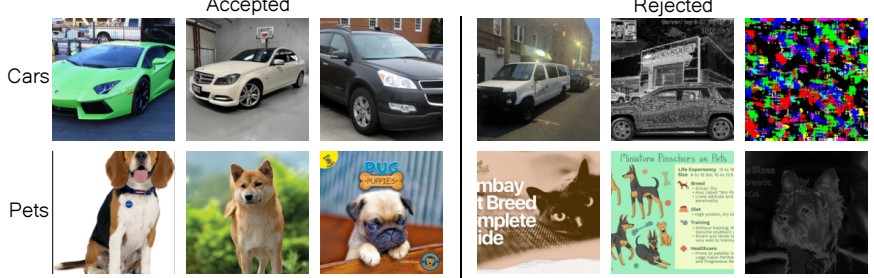

Figure 4: Accepted/rejected examples using our refinement.

backbones, we implemented our work on ViT-B-32, and ResNet-50 alongside ViT-B-16 for a fair comparison. In Table 4, we compare the performance on datasets Flowers, Pets, Food, Cars, and ImageNet.

Using two orders of magnitude fewer data, we outperform IE on the Pets and Food datasets by 0.37% and 1.37%. In the case of Flowers. It should be noted that our zeroshot baseline has a much lower performance (66.22%) than the baseline (94.6%) in Li et al. (2023). Despite this, we obtain a larger margin improvement (24%) than IE (4.5%) on the Flowers dataset which we attribute to our retrieval approach. Compared to SuS-X across all backbones, our approach achieves a consistent improvement over this SOTA approach without any use of image generation models or static datasets such as LAION-5B. Compared to RECO, we observe significant improvements using ViT-B-32 and ResNet-50 across all datasets. Iscen et al. (2023) do not report any results on ViT-B-16.

To identify the source of improvement over the SOTA, we conducted two experiments. (1) We compared retrieving images from the Web with using a static dataset like LAION-5B. Since we lack access to training set labels for uncertain cases, we retrieved images from the most certain ones for CLIP. In this case, we aim to retrieve images similar to certain instances so we can assign the same label to these retrieved images. As shown in Table 5, the accuracy is significantly lower when using static data compared to web data, highlighting the effectiveness

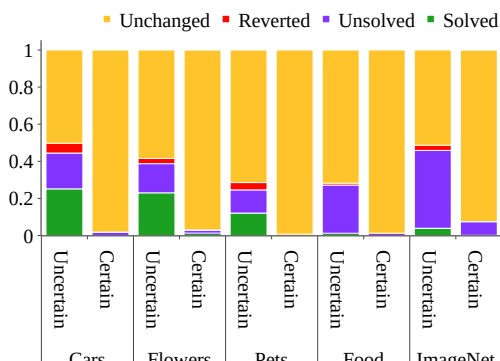

Figure 2: Changes to certainty of instances. **Solved**: Percentage of cases incorrectly predicted before probing and corrected afterwards. **Unsolved**: Incorrectly predicted before and after; **Reverted**: Predicted correctly when using CLIP but wrongly predicted after probing; **Unchanged**: Correctly predicted before fine-tuning and still correctly predicted after.

of search engines for retrieval. (2) To evaluate the impact of linear probing versus more complex methods like Tip-Adapter (Zhang et al., 2021), we present results in Figure K.14. Using Tip-Adapter with $\mathbf{D}_{\text{uncertain}}^{\text{cls}}$ as the retrieved dataset yields higher accuracy than SuS-X (which used an improved Tip-Adapter) on Pets, Cars, Flowers, and Food, and is competitive on ImageNet. This demonstrates the advantage of web data, as our approach consistently improves over the pretrained model regardless of the inference method.

Table 4: Comparison with the state-of-the-art retrieval methods for image classification across 3 different backbones. As can be seen, our approach significantly outperforms the alternatives.

| Backbone | Method | Pets | Cars | Flowers | Food | ImageNet |
|----------|--------|------|------|---------|------|----------|
| **ViT-B-16** | SuS-X | 91.58 | 66.13 | 73.08 | 86.08 | 70.00 |
| | Ours | **93.00** | **82.38** | **86.62** | **88.82** | **70.33** |
| **ViT-B-32** | RECO | - | 68.10 | 56.20 | - | 59.40 |
| | SuS-X | 87.95 | 61.19 | 68.17 | 80.31 | 64.73 |
| | Ours | **90.13** | **75.44** | **83.48** | **84.02** | **65.76** |
| **RN50** | RECO | - | 39.80 | 56.20 | - | 59.40 |
| | SuS-X | 86.35 | 57.14 | 67.32 | 77.02 | 61.66 |
| | IE | 90.80 | - | **99.10** | 84.60 | - |
| | Ours | **91.17** | **69.97** | 90.39 | **85.97** | **62.04** |

As another alternative to LP, we experiment with LoRA (Hu et al., 2022), a parameter-efficient fine-tuning method for transformers. Table 5 shows that LP outperforms CLIP-LoRA on the Flowers and Pets datasets, with only marginal gains on Cars and Food. Despite this, using the retrieved dataset $\mathbf{D}_{\text{uncertain}}^{\text{cls}}$ improves LoRA's performance. This suggests that LoRA's additional parameters may cause overfitting to the retrieved data. Thus, effective data retrieval yields better improvements compared to complex adaptation methods.

## 5.5 Complementarity of datasets

In Table 3, we show (Top) the zeroshot accuracy of CLIP and our method with Linear Probing ($LP$) trained on the best-retrieved dataset (from Tab. 2). Bottom entries show accuracy of these methods when trained on the combination of the training set of each target dataset and the retrieved dataset. We can see our method consistently outperforms ZS CLIP. We observe a significant improvement in Flowers and Cars datasets with over 15% in accuracy, showing that the samples retrieved from the search engine best helps with the features needed to perform well on the test set of these datasets.

On Food and ImageNet, we obtain a slight improvement of 0.24% and 1.81%, respectively. These two are challenging datasets in terms of constructing queries. In the case of Food, we speculate that due to the diversity and variability of the food images online and the specificity of this dataset, the improvements remained marginal. Regarding ImageNet, a significant challenge arises from that certain classes

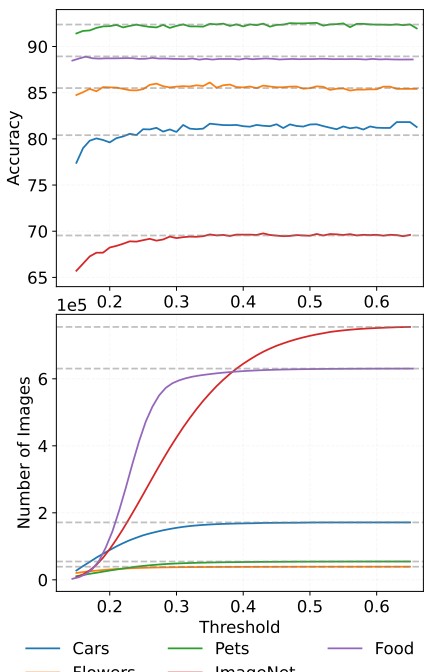

Figure 5: Effect of $\tau_R$ on the number of images and accuracy. Dashed lines represent no refinement.

necessitate the inclusion of highly specialized types of images. Take, for instance, classes representing specific dog species. When attempting to collect images from the Web for these classes, we encounter two issues: a considerable increase in variability within the images or a prevalence of highly specific, limited-content images. Also, we found possible misinterpretations in terms of class names, e.g., the label *Agama* in ImageNet, could result in images related to the Indian religion or a lizard.

For Pets, we obtain a significant improvement of more than 3% of accuracy, surpassing the performance of training a Linear Probe on the training set of the target dataset. When we combined our retrieved dataset with the target dataset's training set, we observed consistent improvements over baseline methods, indicating that the retrieved images supplement the dataset's training distribution. Figure 2 shows a more detailed comparison of CLIP predictions before and after update using retrieved images. In this bar plot, we can see that the slight improvement in the Food dataset is due to the small margin in the reverted cases relative to the solved cases. This is while reverted cases make up a much smaller percentage of the changes in the uncertain cases of other datasets. Moreover, the effect of our method is minimal in certain cases where the percentage of unchanged cases makes up the largest fraction of every dataset. In other words, adding a

Table 5: Performance of LP vs **LoRA** on $\mathbf{D}^{\texttt{cls}}_{\text{uncertain}}$ and using LAION-5B for retrieval.

|  | Flowers | Pets | Cars | Food |
|---|---|---|---|---|
| *ZS ViT-B-16* | 71.15 | 89.04 | 64.71 | 88.73 |
| *Using LAION-5B* | 61.86 ↓-9.29 | 80.84 ↓-8.20 | 56.60 ↓-8.10 | 88.35 ↓-0.38 |
| *LP w/ $\mathbf{D}^{cls}_{uncertain}$* | 86.24 ↑15.09 | 92.61 ↑3.57 | 80.52 ↑15.81 | 88.97 ↑0.24 |
| *LoRA w/ $\mathbf{D}^{cls}_{uncertain}$* | 84.29 ↑13.14 | 92.56 ↑3.52 | 80.55 ↑15.84 | 88.98 ↑0.25 |

Linear Probe to deal with uncertain instances does not affect the instances where CLIP is certain about the prediction.

## 5.6 On the refinement process

**The value of threshold $\tau_R$.** Figure 5 shows the effect of a strict to a more relaxed refinement threshold $\tau_R$ on the accuracy of our refinement process in the retrieved dataset $\mathbf{D}^{\texttt{cap}}_{\text{uncertain}}$. Notice that small values in $\tau_R$ mean we reduce the acceptance range (see section 4.3). We use a threshold of 0.45 across all the datasets, where we observe performance improvement in a variety of benchmarks whilst not removing too large a portion of the retrieved dataset, thus covering the target dataset distribution (see section 7).
**Qualitative analysis of refinement**. Figure 4 shows random samples of our refinement process's accepted and rejected instances for Pets and Food datasets. Other datasets are in Appendix L. The refinement process can reject images that contain words related to the class names on the dataset but do not necessarily contain the object of interest, thus not providing any useful information.

## 5.7 On the value of the threshold $\tau_H$

To examine the effects of the Shannon confidence threshold $\tau_H$ on uncertainty quantification (4.1), we perform experiments on the target dataset's training set. Figure 3 (c) shows a correlation between the number of incorrectly predicted samples considered uncertain at a given $\tau_H$ and the total incorrect predictions. This correlation highlights the effectiveness of using uncertainty as a proxy for identifying incorrect predictions, consistently observed across all datasets.

Figure 3 (b) shows the percentage of uncertain classes as a factor of the threshold $\tau_H$. For the Food and ImageNet datasets, uncertainty approaches 100% indicating at least one uncertain instance per class across thresholds. In contrast, other datasets show a gradual increase in uncertainty, suggesting the model is more confident about some classes. To maximize the capture of incorrectly predicted samples using uncertainty, selecting a threshold value near 0.9 would achieve this objective.

To better understand the impact of uncertainty-aware retrieval, we conduct an ablation in which retrieval is performed for all classes in each dataset, regardless of the model's uncertainty. This setup serves as a baseline for evaluating the benefit of skipping retrieval when the model is confident in its predictions. The results of this experiment are presented in Appendix C. We observe that removing uncertainty from the retrieval process consistently degrades performance across datasets, with an average accuracy drop of 4.26%. Notably, on ImageNet and Food101, performance in this baseline falls below that of the Zero-Shot setting, highlighting the importance of using uncertainty in the retrieval pipeline.

## 6 Data Leak

A key concern with using search engines to build datasets is the risk of performance gains being driven by retrieving test set images. Here we assess the extent to which the retrieved images match those found in the test set of the target dataset by employing a hash comparison method. We use Difference Hashing (dHash) (Buchner, 2023) to evaluate the similarity between images. This method first converts each image in the target and retrieved sets to its byte representation and hashes it, resulting in a unique identifier for the individual content of each image. The number of overlapping hash codes, serving as a quantitative measure of data leakage between the datasets, are reported in Table 6. Our method retrieves a minuscule portion

Table 6: Number and percentage of leaked images of the test set.

|  | Flowers | Pets | Cars | Food | ImageNet |
|---|---|---|---|---|---|
| Target test set size | 6148 | 3669 | 8041 | 25250 | 50000 |
| $\mathbf{D}^{cls}_{uncertain}$ | 5 ($\approx$ 0.0%) | 2 ($\approx$ 0.0%) | 105 (1.3%) | 0 (0.0%) | 107 ($\approx$ 0.0%) |
| $\mathbf{D}^{cap}_{uncertain}$ | 4 ($\approx$ 0.0%) | 7 ($\approx$ 0.0%) | 448 (5.5%) | 0 (0.0%) | 121 ($\approx$ 0.0%) |
| $\mathbf{D}^{desc}_{uncertain}$ | 5 ($\approx$ 0.0%) | 17 ($\approx$ 0.0%) | 245 (3.0%) | 0 (0.0%) | 184 ($\approx$ 0.0%) |
| Accuracy Drop (%) | 0.01 | 0.00 | 0.08 | —— | 0.00 |

of test set images online, thus no leakage. Therefore, the substantial consistent improvement resulting from our approach cannot be attributed to test set leakage, implying that its enhanced performance stems from improved feature learning and generalization.

## 7 Effects of Refinement Threshold on Retrieved and Target Datasets

Figure 6 shows UMAP visualizations of the image features from the Cars dataset and the retrieved dataset for two threshold values $\tau_R$. The retrieved data aligns well with the training distribution and fills class-specific gaps, explaining the performance gains of our Linear Probe in uncertain cases. Moreover, a stricter threshold $\tau_R$ retains fewer instances in the retrieved set. A balanced compromise can be achieved by selecting a retrieved dataset that best overlaps the target distribution while excluding noisy samples.

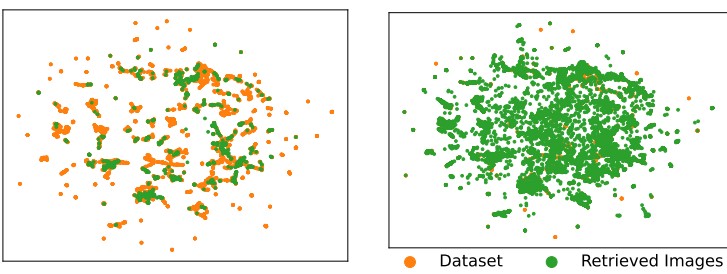

Figure 6: UMAP visualization of the distribution of the image features from ViT-B-16 for the training split of the original and retrieved dataset. It shows the effect of the choice of the threshold $\tau_R$ refinement method on the retrieved dataset from a less (right) to a more strict (left) refinement.

## 8 On the Number of Images per Query

To study the effect of the number of images retrieved per class, we train a classifier on subsets of increasing size from $\mathbf{D}^{cls}_{uncertain}$ and report the accuracy results for each dataset in Figure 7. For Cars, Flowers and Pets we observe a steady increase in accuracy as the number of images from the retrieved set used for training increases. This is while the lowest number of used images (25 for all datasets) still outperforms the zero-shot accuracy for every dataset except Food. In ImageNet, on the other hand, we observe a slight decline in accuracy when using 50 images, but the overall trend is upward with more images.

Furthermore, in Cars, Flowers, and Pets datasets, the rate of accuracy improvement in relation to the increase in the number of images diminishes as more data is incorporated, eventually reaching a plateau at around 100 images. This phenomenon arises from insufficient additional information in newer images or the limitations of the search engine in retrieving more relevant images for the classifier.

## 9 Conclusion

The proposed method of using search engines for machine learning is innovative and intersects with areas like active learning, weakly supervised learning, cross-modal retrieval, and language models. In the future, we envision pre-trained models dynamically accessing the Internet to acquire new knowledge and handle

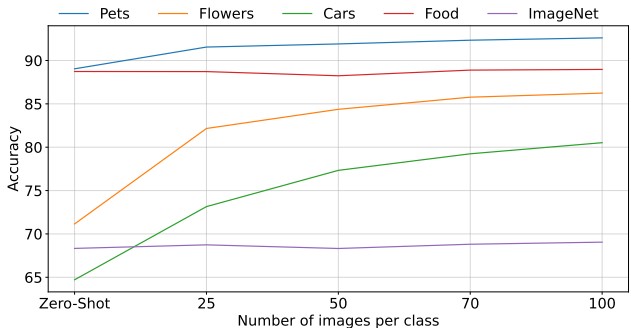

Figure 7: Exploring the impact of varying the number of images utilized in training a LinearProbe.

novel concepts as they arise using constantly refined datasets from varying sources, such as large-scale uncurated datasets and multiple, potentially retrieval-specific search engines. This contrasts with traditional static models, which are designed to handle unseen instances, a challenge that current out-of-distribution generalization research aims to overcome. Similar to effective prompt engineering, our approach suggests that crafting well-constructed search queries (potentially with LLMs) could enhance performance, warranting further exploration. It is worth noting that our approach may face limitations in resource-constrained areas, like specialized medical fields, where purpose-built search engines might be more effective. One avenue for potential future research involves amalgamating results from various search engines. This could lead to divergent sets of retrieved data. Additionally, the susceptibility to bias is a pertinent concern with such datasets. As these methods mature, addressing such issues becomes increasingly important.

## 10    Limitations

While effective, our method has several limitations. It relies on publicly accessible web data, limiting applicability in **specialized domains** like medical imaging, where relevant data is often private, though users in such fields may have access to suitable datasets. The method also struggles with entirely novel or **out-of-distribution** concepts, as retrieval cannot help when pre-trained representations are lacking. This highlights the importance of pre-training coverage. **Search engine variability** across time, region, personalization, or algorithm updates arbitrarily affecting search result ranking can affect both reproducibility and stability. **Ambiguous class names** (e.g., "Agama") can further degrade retrieval quality by introducing unrelated content. Finally, although more efficient than many alternatives, our method still adds inference-time **overhead from retrieval**, refinement, and adaptation, which may be unsuitable for resource-limited settings.

## 11    Broader Impact

Our method relies on publicly available web data, raising important ethical and legal considerations:

**Copyright and IP.** Many web images and texts are copyrighted, and using them without permission, especially commercially, risks IP infringement. Ongoing cases, like those involving OpenAI, will shape the legality of using public data for training and retrieval (Spanglett, 2025).

**Web Scraping Ethics.** Responsible scraping requires honoring protocols like `robots.txt`, identifying crawlers, and respecting opt-out signals. Bypassing such protections breaks ethical and legal boundaries.

**Bias in Retrieved Data.** Since our method relies on web search engines, it inherits existing societal biases present in online content. These can manifest in how certain groups or concepts are represented, potentially reinforcing harmful stereotypes. Addressing this remains an important direction for future work through techniques such as bias-aware retrieval, filtering, or balancing strategies.

**Environmental Impact.** While less resource-intensive than large-scale training, our approach adds overhead from search, storage, and fine-tuning, contributing to carbon emissions at scale. Future work should reduce this footprint by reusing data or improving refinement efficiency.

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

# A  Information-Theoretic Justification for Uncertainty-Based Retrieval

We provide theoretical justification for uncertainty-based retrieval and its effectiveness by analyzing the information gain properties of uncertain instances. Let $X$ denote the set of input images, $Y$ the set of possible class labels, and $x \in X$ a specific input instance. We denote $\mathbb{H}(Y|x)$ as the predictive entropy of the pre-trained model for instance $x$, measuring the uncertainty in label prediction. Let $X_{\text{retrieved}}$ represent the set of images retrieved from the search engine based on instance $x$. The information gain $\text{IG}(Y; X_{\text{retrieved}}|x)$ quantifies how much the retrieved data reduces uncertainty about the true label for instance $x$.

**Proposition 1** (Information Gain Monotonicity). *For instances $x_1, x_2 \in X$ where $\mathbb{H}(Y|x_1) > \mathbb{H}(Y|x_2)$, the expected information gain from retrieved data satisfies:*

$$\mathbb{E}[IG(Y; X_{retrieved}|x_1)] \geq \mathbb{E}[IG(Y; X_{retrieved}|x_2)] \tag{9}$$

*where $IG(Y; X_{retrieved}|x) = \mathbb{H}(Y|x) - \mathbb{H}(Y|x, X_{retrieved})$ is the information gain, with $\mathbb{H}(Y|x)$ being the initial predictive entropy and $\mathbb{H}(Y|x, X_{retrieved})$ the entropy after incorporating retrieved data.*

*Proof.* We prove this by establishing the relationship between initial entropy and maximum achievable information gain. By definition of conditional entropy and its non-negativity property, we have:

$$\mathbb{H}(Y|x, X_{\text{retrieved}}) \geq 0 \tag{10}$$

Therefore, the information gain is bounded by the initial entropy:

$$\text{IG}(Y; X_{\text{retrieved}}|x) = \mathbb{H}(Y|x) - \mathbb{H}(Y|x, X_{\text{retrieved}}) \leq \mathbb{H}(Y|x) \tag{11}$$

The maximum possible information gain occurs when uncertainty is completely resolved, i.e., when $\mathbb{H}(Y|x, X_{\text{retrieved}}) = 0$:

$$\max \text{IG}(Y; X_{\text{retrieved}}|x) = \mathbb{H}(Y|x) - 0 = \mathbb{H}(Y|x) \tag{12}$$

Since the maximum achievable information gain equals the initial entropy, and $H(Y|x_1) > H(Y|x_2)$ by assumption, we have:

$$\max \mathbb{E}[\text{IG}(Y; X_{\text{retrieved}}|x_1)] = \mathbb{H}(Y|x_1) > H(Y|x_2) \tag{13}$$

$$= \max \mathbb{E}[\text{IG}(Y; X_{\text{retrieved}}|x_2)] \tag{14}$$

Under the reasonable assumption that retrieved data quality is comparable across instances (search engines return similarly relevant results for different queries), the expected information gain maintains the same monotonic ordering as the maximum achievable gain. Therefore:

$$\mathbb{E}[\text{IG}(Y; X_{\text{retrieved}}|x_1)] \geq \mathbb{E}[\text{IG}(Y; X_{\text{retrieved}}|x_2)] \tag{15}$$

where $\mathbb{E}[\cdot]$ denotes expectation over the randomness in the retrieval process.  □

**Corollary 1** (Effect of Uncertainty-Based Selection). *Given a fixed budget $B$ for data retrieval, selecting instances with the highest predictive entropy $\mathbb{H}(Y|x)$ maximizes the expected total information gain.*

*Proof.* Let $X_{\text{uncertain}} = \{x \in X : \mathbb{H}(Y|x) \geq \tau\}$ and $X_{\text{certain}} = \{x \in X : \mathbb{H}(Y|x) < \tau\}$ for some threshold $\tau > 0$, where $X_{\text{uncertain}}$ represents the set of uncertain instances and $X_{\text{certain}}$ the set of certain instances. Given budget constraint $|X_{\text{selected}}| \leq B$, the total expected information gain is:

$$\mathbb{E}[\text{IG}_{\text{total}}] = \sum_{x \in X_{\text{selected}}} \mathbb{E}[\text{IG}(Y; X_{\text{retrieved}}|x)] \tag{16}$$

$$\geq \sum_{x \in X_{\text{uncertain}} \cap X_{\text{selected}}} \mathbb{E}[\text{IG}(Y; X_{\text{retrieved}}|x)] \tag{17}$$

by Theorem 1, where $X_{\text{selected}} \subseteq X$ is the set of instances chosen for retrieval. The inequality becomes equality when $X_{\text{selected}} = X_{\text{uncertain}}$ (up to the budget constraint), demonstrating the optimality of uncertainty-based selection.  □

Therefore, instances with higher predictive entropy $\mathbb{H}(Y|x)$ provide greater expected information gain $\mathbb{E}[\text{IG}(Y; X_{\text{retrieved}}|x)]$, making them optimal targets for search engine retrieval. Considering the information-based prediction mechanism in Eq. (8), we also specifically target instances that empirically show information gain given the predictions from the fine-tuned linear classifier. This is in line with the justification above, where we only seek predictions based on the new retrieved information only if the prediction provides better confidence compared to the zero-shot model. Furthermore, this explains why retrieving data for already-certain instances (low $\mathbb{H}(Y|x)$) provides minimal benefit. Their low entropy leaves little room for information gain, as the upper bound $\text{IG}(Y; X_{\text{retrieved}}|x) \leq \mathbb{H}(Y|x)$ is already small. This is further validated by our experiments in Appendix C, which show that using certain labels leads to performance degradation due to the linear classifier outputting overconfident distributions.

## B  Effects of prompt specificity on prediction confidence

Since predictions from CLIP are reliant not only on the input images but also on the text prompt, the uncertainty of the prediction will also be affected by this text. To study the effects of the specificity of the input prompt on the confidence in the prediction, we use six methods of prompting, each at different levels of descriptiveness. The results of this study are visualized in Figure B.8.

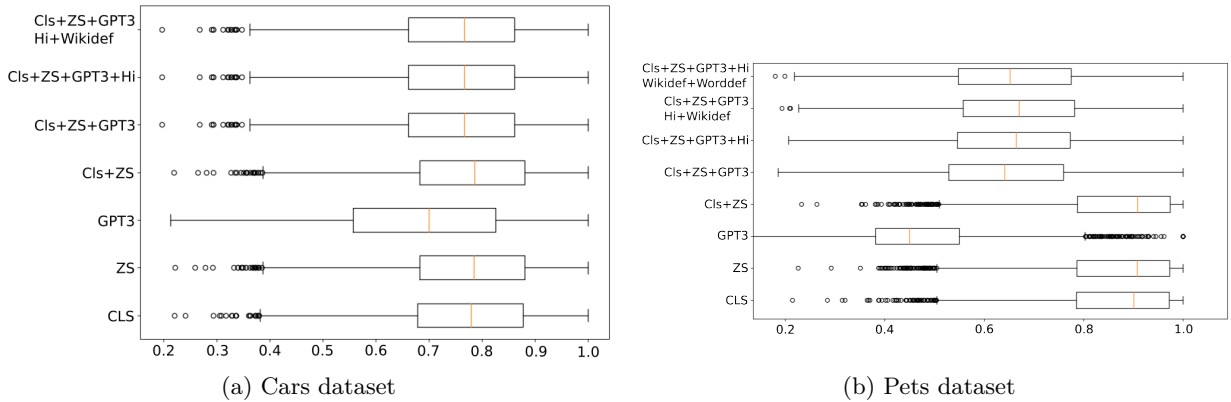

(a) Cars dataset                                        (b) Pets dataset

Figure B.8: Boxplot of confidence values when prompts with varying specificity are used. **CLS** refers to using class names from the dataset for prompting. **ZS** are prompts specific to each dataset described in more detail in Appendix B. **GPT3**, **Wikidef**, **Hi** and **Worddef** are prompt templates taken from Li et al. (2022a). **GPT3** refers to descriptions of each class generated by OpenAI-GPT3 (Zhao et al., 2023) and used as the prompt. **Worddef** and **Wikidef** are Wordnet (Miller, 1998) and Wiktionary (Wiktionary Contributors, 2023) definitions of each class in the dataset (With the exception of Cars for which no classes have Wordnet definitions). **Hi** refers to a set of words extracted from Wordnet based on the ontological hierarchy of the class name.

The prompt named CLS refers to using class names with the format "`{class_name}`" to prompt CLIP. ZS expands this approach by adding a generic description of the dataset in the format "`{class_name}, a type of {description}`", e.g. the prompt used for Pets is "`{class_name}, a type of pet`". For GPT3, a description of the class name is generated by OpenAI GPT3 first, and the description is used for prompting without the addition of the class name that was used to generate the description or any other phrases. Wordnet and Wikidef refer to the definition of the class name in WordNet (Miller, 1998) and Wiktionary (Wiktionary Contributors, 2023). Similar to GPT3, no additional phrases were added for prompting. Finally, Hi refers to generating a prompt using the WordNet hierarchy where the last three words in the ontological tree leading to the class name are concatenated and used as the prompt.

Figure B.8 indicates that utilizing ZS individually, or in combination with CLS, marginally enhances the network's confidence in predictions. This enhancement is attributable to a more refined representation of the prompts, thereby reducing text ambiguity. A notable observation is the significant reduction in

network confidence with the use of GPT-3 descriptions compared to simpler prompts such as CLS or ZS. An examination of these descriptions revealed that GPT-3 prompts tend to be more narrative and detailed rather than focusing on keyword specificity. That is, instead of representing a concise combination of the class's crucial characteristics, these prompts offer extensive narratives, often overlooking the visually distinguishable aspects of the class.

## C  Ablation on usage of uncertainty-aware retrieval mechanism

To assess the importance of our uncertainty in our retrieval mechanism, we remove this component and instead retrieve images for all classes present in the dataset and call this method Baseline. Table C.7 shows the performance obtained for each dataset compared to our uncertainty-aware approach.

Table C.7: Comparison of Zero-shot, Our Approach, and Baseline on multiple datasets.

|  | ImageNet | Flowers | Cars | Pets | Food101 |
|---|---|---|---|---|---|
| Zero-shot | 68.33 | 71.15 | 64.71 | 89.04 | 88.73 |
| Baseline | 57.93 | 85.79 | 80.45 | 90.79 | 81.13 |
| Our Approach | **69.05** | **86.24** | **80.52** | **92.61** | **88.97** |

Performance on ImageNet and Food101 severely degrades when retrieving images from a search engine without regard to uncertainty. As mentioned in section 5.2, classes present in Food101 have low specificity, i.e., for each label, the distribution of images belonging to that class is broad. This makes it difficult to retrieve images that closely match the distribution of the dataset from the Web. This was anticipated before conducting any experiments, based on the results in Table 1. ImageNet, on the other hand, consists of a large number of classes (1000 in total), and without refinement and uncertainty-focused retrieval and prediction, the trained probe encounters significant levels of noise, degrading its performance.

For Pets and Flowers, we observe slight performance degradation due to disregarding the classes the model is certain about. Performance on Cars, however, is largely unaffected. This indicates that when labels are highly specific (e.g., Toyota Corolla 2012 Sedan), the retrieved images are adequately representative of the dataset. Moreover, due to the nature of this dataset, high-quality images are easy to find online.

## D  Refinement using on-manifold clustering

Figure D.9 shows a visualization of the clustering on the hypersphere using von Mises-Fisher distributions. The images from the dataset may be used for forming clusters where the colormap of the manifold in Figure D.9 visualizes the probability density. We reject features corresponding to retrieved images that are too far off from the clusters by setting a threshold across datasets. For cases where the number of clusters and images are too large (such as ImageNet), we initialize the centers of the clusters using the normalized outputs of a Riemannian k-means algorithm for faster convergence.

## E  Other refinement methods

The refinement method described in section 4.3 shows the exclusion of unrelated and noisy images from the retrieved set; however, this method is intrinsically dependent on the training split of the target dataset. In this experiment, we investigate an alternative refinement approach that is not dependent on accessing images from the dataset and explore an approach that circumvents the computationally demanding clustering strategy depicted in section 4.3.

For refining retrieved images, we need anchor points that facilitate gauging the similarity of each retrieved image to the target data distribution. Even without access to images from the target dataset, textual values of labels for the target dataset can still serve as a viable alternative. Essentially, this process involves conducting inference on the retrieved images, identifying text with the highest similarity to each image,

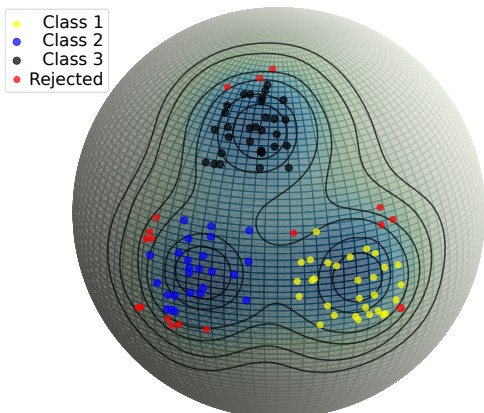

Figure D.9: Conceptual visualisation of CLIP features. Dots represent images from 3 classes and the colormap indicates probability density from a mixture of VMF distributions. Threshold lines demonstrate selection boundaries, with images beyond these lines (in red) rejected during refinement.

and discarding those images with their similarity value falls below a predefined threshold, denoted as $\tau_R$ according to the following equation:

$$f(\boldsymbol{x}_r \mid \boldsymbol{y}) \quad = \quad \mathbb{I}\big[\langle \phi_x(\boldsymbol{x}_r), \phi_t(\boldsymbol{y}_{i^*}) \rangle < \tau_R\big], \tag{18}$$

$$i^* \quad = \quad \arg\max_i \langle \phi_x(\boldsymbol{x}_r), \phi_t(\boldsymbol{y}_{i^*}) \rangle. \tag{19}$$

On the other hand, as a substitute for the clustering on the unit hypersphere methodology detailed in section 4.3, we investigate the application of the more efficient k-means clustering. Recognizing that k-means is not inherently designed for clustering on a sphere, we start by elevating each point on the hypersphere to the tangent space at the mean of the image modality. This mean is obtained using the images from the training split of the target dataset. Consequently, the overall procedure for this strategy is similar to the one described in section 4.3 with the exception of employing the Euclidean tangent space, thereby facilitating clustering using k-means. The outcome of the text-based approach is visualized in Figure E.10 and for the k-means approach in Figure E.11.

While the approach proposed in section 4.3 gives better results than the ones seen in Figure E.10 and E.11, we still see some improvement especially for Cars in both figures. Two observed issues with the tangent space clustering are the issues with the distortion in the projection of images to the tangent space and the other is the problems with failure of k-means in finding appropriate cluster centers leading to large decline in the accuracy observed in Figure E.11a.

## F   Retrieval details

**Timing**: Our retrieval process for Pets, Flowers, Food, and Cars took less than 1 hour for each dataset. For ImageNet, this process takes less than a day using a single core process. By parallelizing retrieval, depending on API quotas, this process can be finished for ImageNet in less than 1 hour as well.

**Costs**: Pricing is API and time dependent and these APIs are provided in a pay-as-you-go model. At the time of our experiments Google pricing was set at 4.50 USD per 1000 requests up to 5 million requests.

## G   Image distributions comparison

To qualitatively investigate the differences between the distribution of images from the target dataset to that of the retrieved dataset, we train a classifier to detect which dataset the images are coming from. In particular, analogous to a discriminator in a generative adversarial network (GAN), we train a binary classifier to distinguish the retrieved dataset from the target one. The outputs of this classifier are shown in

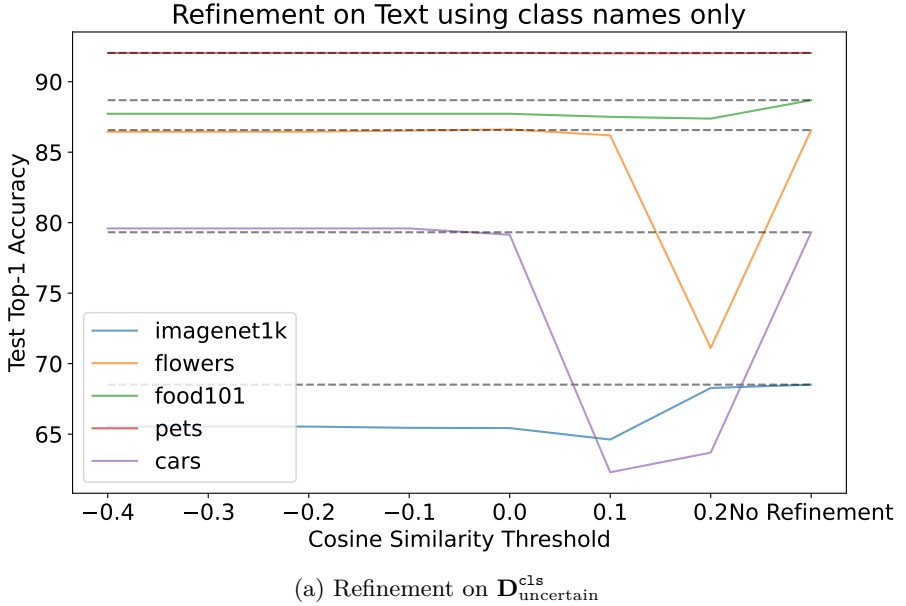

(a) Refinement on $\mathbf{D}^{\text{cls}}_{\text{uncertain}}$

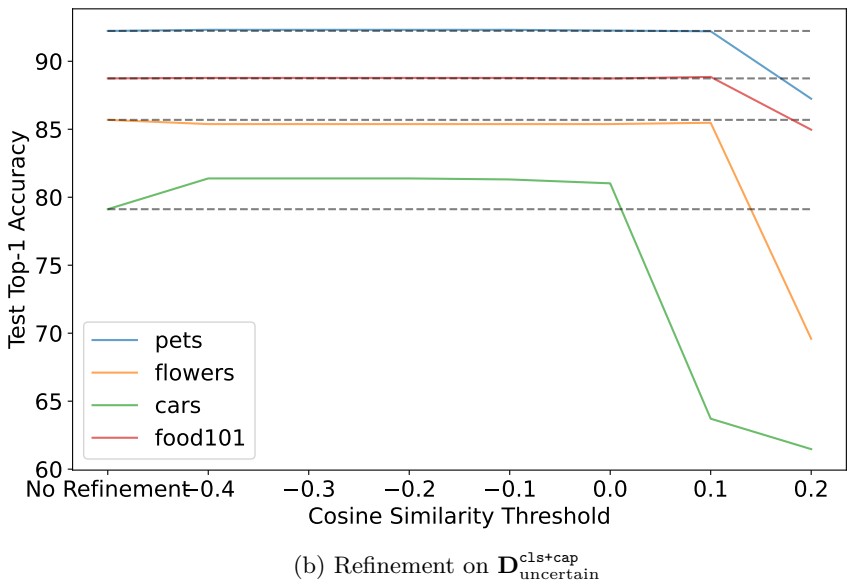

(b) Refinement on $\mathbf{D}^{\text{cls+cap}}_{\text{uncertain}}$

Figure E.10: Refinement using the labels of the target dataset

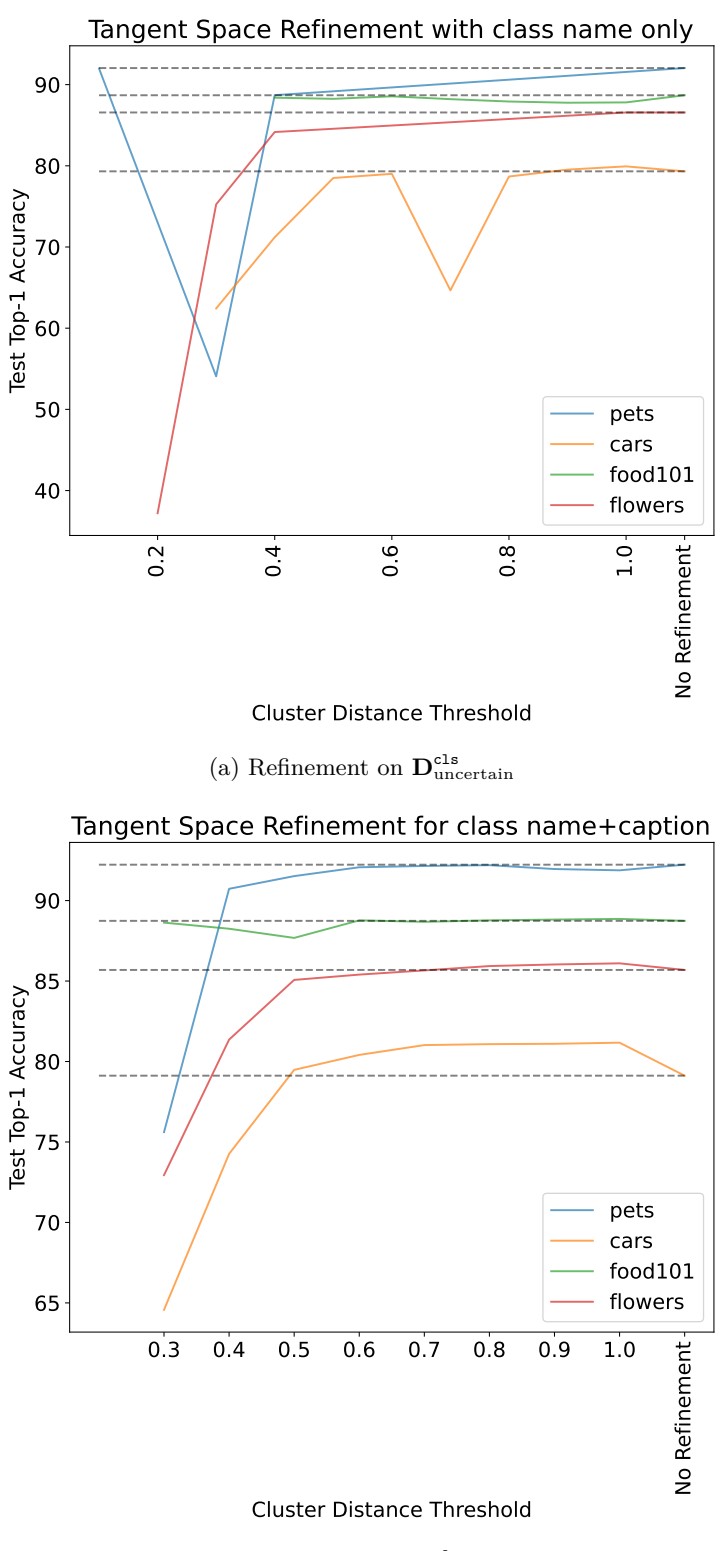

(a) Refinement on $\mathbf{D}^{\mathbf{cls}}_{\mathrm{uncertain}}$

(b) Refinement on $\mathbf{D}^{\mathbf{cls+cap}}_{\mathrm{uncertain}}$

Figure E.11: Refinement using the tangent space of the image modality with k-means clustering

Figure G.12 for different datasets. We anticipate that instances characterized by increased overlap–indicative of greater confusion by the classifier–will exhibit a more closely aligned distribution. For the Food and Pets dataset, there is minimal overlap, suggesting that the retrieved data exhibits less similarity.

We hypothesize that this easy distinction by the classifier is because of a combination of reasons, one of which is a genuine shift between the distribution of retrieved and target images. This could be due to the lack of specificity in the class names used to retrieve images. Consider the scenario of retrieving images of cars produced by a specific manufacturer. In this case, images of the same cars, sharing the same make and model, from both the retrieved and target datasets may exhibit little visual distinction. This is attributed to their common manufacturer and the descriptive nature of the class names, which facilitates a close match between the images in the two datasets. On the other hand, when it comes to the Food or Pets dataset, images retrieved by a search engine will be much more diverse than their corresponding curated ones, which indicates biased distributions. Therefore, distribution shifts are expected for these datasets. The Flowers dataset seems to have a good overlap while retrieving many distinct images.

Another potential reason for this distinction could be much shallower than the previously suggested one. The images from the target dataset are curated images that commonly have a fixed height-to-width ratio. On the other hand, retrieved images will have different ratios even among themselves. While resizing and cropping images before inputting them into CLIP, there's a potential for consistent content cropping in the retrieved images, whereas the entirety of the object's content in the target dataset remains intact. This could serve as a superficial cue for the classifier, potentially leading to shortcut learning.

## H  Quantitative measure of difference between retrieved and target datasets

To quantify the difference between the distribution of features of the retrieved images compared to that of the target dataset, we use the Earth Mover's Distance (EMD) (Rubner et al., 2000). EMD or the Wasserstein distance between the two distributions can show how different the retrieved images are in the feature space compared to the target dataset by quantifying the minimum cost of turning one distribution into the other, where cost is measured in terms of moving mass through the space. The value for Pets, Cars, Flowers and Food datasets is visualized in Figure H.13. Comparing the datasets to each other, Flowers has the lowest difference between the retrieved and target dataset while Pets has the highest difference. When crossing these values with the results in Table 3, no correlations are observed between the EMD and obtained increase in accuracy given that the EMD for the Food dataset lies between datasets that obtain larger improvements. However, when comparing these results with Figure 6, it is evident that the retrieved images are mostly close to the clusters formed by the images from the target dataset. This implies that although the Wasserstein distance between the retrieved and target datasets is low, the observed increase in accuracy may be attributed to the retrieval of images situated at the tails of the distribution, particularly notable in datasets like Food, which feature a diverse array of images.

## I  Ablations on CLIP backbones

We perform ablations on using different CLIP backbones with our retrieved datasets $\mathbf{D}^{\texttt{cls}}_{\text{uncertain}}$, i.e., ViT-g-14 and ViT-L-14. Consistently achieving enhanced zero-shot classification performance across these datasets, with the exception of Food. Notably, in the case of ViT-L-14, Food is already achieving an accuracy level of 97%. The primary factor hindering improvement in this instance is that CLIP ViT-L-14 already encompasses a substantial portion of the Food dataset's distribution. Consequently, our retrieved set does not contribute additional information, see Table I.8.

## J  Analysis of using pre-trained model vs updated model using the retrieved set

Once the classifier is trained on the retrieved images, the decision to use either the classifier or CLIP itself during inference will be determined by Eq. (8), as detailed in the paper. Table J.9 provides the proportion of instances the LP classifier is used. Datasets with the largest accuracy gains, namely Flowers and Cars, pass the largest proportion of images through the classifier. Conversely, Food101 showed minimal improvement

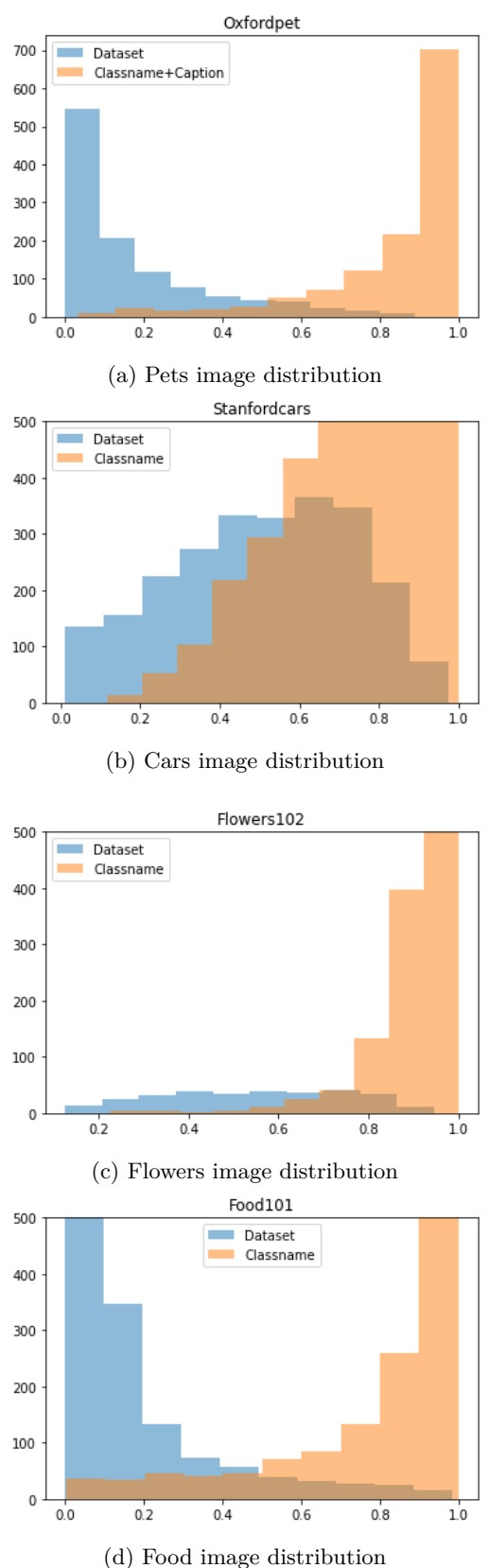

(a) Pets image distribution

(b) Cars image distribution

(c) Flowers image distribution

(d) Food image distribution

Figure G.12: Distribution of the retrieved images compared to the images from the target dataset.

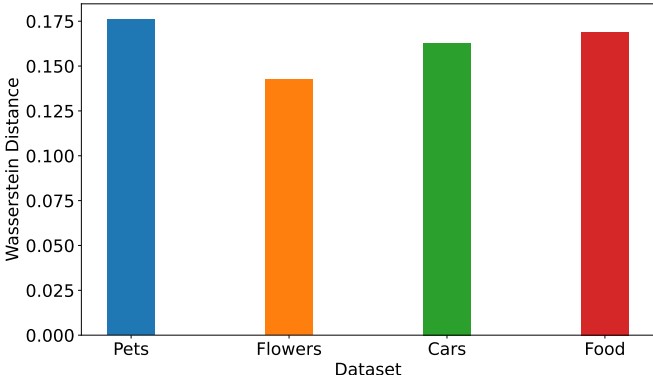

Figure H.13: EMD between the feature distribution of the retrieved and target images

Table I.8: Performance of different CLIP backbones by using our method. We can see a consistent performance improvement in each of the datasets.

|  | Flowers | Pets | Cars | Food | ImageNet |
|---|---|---|---|---|---|
| *Zero Shot CLIP ViT-g-14 LAION-2B* | 77.35 | 94.33 | 92.92 | 91.55 | 76.64 |
| Ours | 86.53 ↑9.18 | 94.41 ↑0.81 | 93.73 ↑0.81 | 91.61 ↑0.06 | 77.12 ↑0.48 |
| *Zero Shot CLIP ViT-L-14 OpenAI* | 79.17 | 93.43 | 77.94 | 97.36 | 75.53 |
| Ours | 87.88 ↑8.71 | 94.28 ↑0.85 | 85.76 ↑7.82 | 97.24 ↓0.12 | 76.06 ↑0.53 |

and the lowest classifier utilization rate correspondingly. This shows the effectiveness of the entropy-driven classifier selection method: better retrieved images for Flowers and Cars led to lower entropy in the classifier's decisions, enhancing confidence in its predictions relative to CLIP. Conversely, the suboptimal image retrieval for Food101 leads to increased entropy and diminishes the reliability of classifier predictions in comparison to CLIP's outcomes.

Table J.9: Percentage of instances from each split of a dataset passed through the classifier as opposed to CLIP according to Eq. (8).

|  | Pets | Flowers | Cars | Food101 | ImageNet |
|---|---|---|---|---|---|
| Train | 76.30 | 89.11 | 87.90 | 65.94 | 80.47 |
| Test | 75.87 | 89.44 | 87.69 | 65.95 | 80.74 |

## K   Using our method for few-shot learning

Tip-Adapter (Zhang et al., 2021) is an adaptation technique designed for CLIP, enabling few-shot classification without extensive training. It inherits the training-free advantage observed in zero-shot CLIP, while simultaneously achieving competitive performance in comparison to methods requiring extensive training. Tip-Adapter constructs an adapter using a key-value cache model based on the few-shot training set. Moreover, by fine-tuning the cache model, Tip-Adapter can achieve state-of-the-art (SOTA) performance with significantly fewer training epochs than existing methods, demonstrating both effectiveness and efficiency. We chose this method to test our dataset since it is a training-free technique that can show how relevant it is to have a dataset that matches the distribution of the target dataset. We performed experiments using Tip-Adapter with the retrieved dataset $\mathbf{D}^{\mathtt{cls}}_{\mathrm{uncertain}}$. These experiments are performed 100 times per dataset with different seeds. Thus, using different images for each k-shot. We use $k = 2, 4, 8.16$ instances from the retrieved dataset. Figure K.14 shows the performance in each benchmark. We noted that both variants of

the Tip-Adapter (Training-Free and Fine-Tuning) derived benefits from our retrieved dataset. This underscores the effectiveness of our methodology in automatically constructing a dataset that comprehensively represents the distribution of the target dataset, extending beyond just our linear probing approach.

## L    Qualitative analysis of the refinement

To give a more intuitive sense of what images are rejected or accepted during the refinement process we provide the visualizations in Figure L.15, Figure L.16 and Figure L.17. These visualizations show random samples from the retrieved dataset that were accepted/rejected by the refinement procedure. A common phenomenon in the retrieval process is the acquisition of images that merely match the textual description of the queried class name but lack any shared visual similarity. As the CLIP features are employed in the refinement procedure, we effectively filter out unrelated images by applying a threshold to the similarity between the features of the retrieved and target datasets.

## M    Accuracy improvement of top classes

To study the changes in the accuracy of each dataset on a more granular level, we quantify the changes in the accuracy of each class. Then, we visualize the top-10 and bottom-10 classes according to these values in Figure M.18, Figure M.19, Figure M.20, Figure M.21 and Figure M.22. We also provide visualizations of the confidence for each of the top and bottom classes in the corresponding figure. In general, the bottom-10 classes exhibit higher confidence (lower entropy), meaning that in the case of $\mathbf{D}^{\mathtt{cls}}_{\mathrm{uncertain}}$, either no images are retrieved for that class or using Eq. 8, we rely mainly on CLIP for these classes. Meanwhile, in the case of $\mathbf{D}^{\mathtt{cap}}_{\mathrm{uncertain}}$ due to the instance-based retrieval method that generates a query per uncertain image, more images could be retrieved for classes for which CLIP has a lower overall confidence.

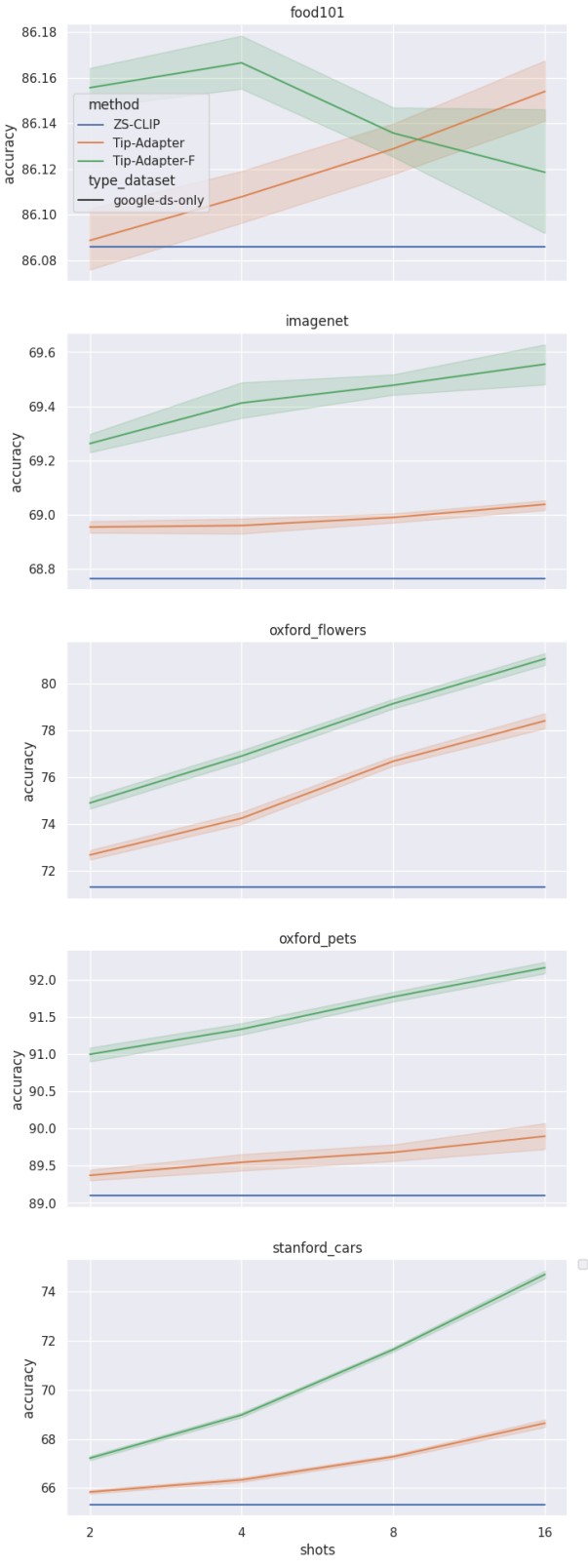

Figure K.14: Few-shot experiments using Tip-Adapter (Zhang et al., 2021) with our retrieved dataset $\mathbf{D}_{\text{uncertain}}^{\text{cls}}$

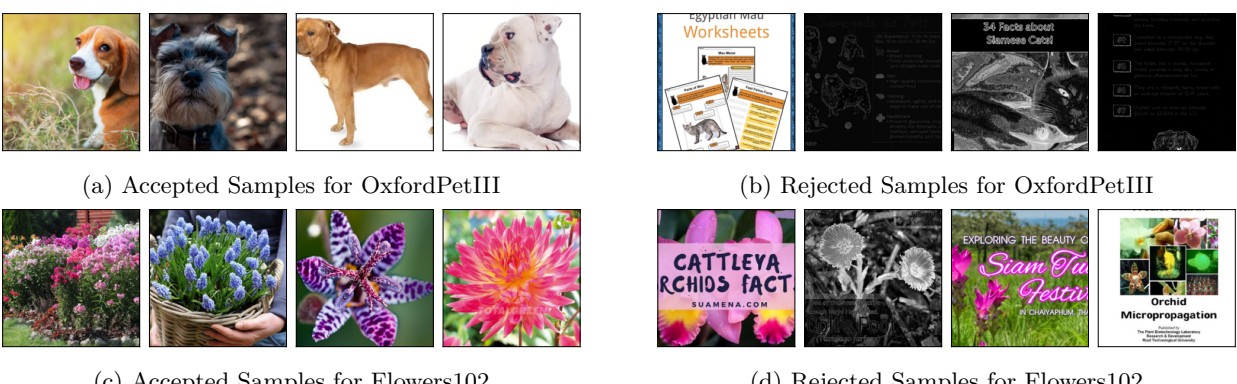

(a) Accepted Samples for OxfordPetIII

(b) Rejected Samples for OxfordPetIII

(c) Accepted Samples for Flowers102

(d) Rejected Samples for Flowers102

Figure L.15: Random samples from each retrieved dataset based on the refinement procedure described in section 4.3

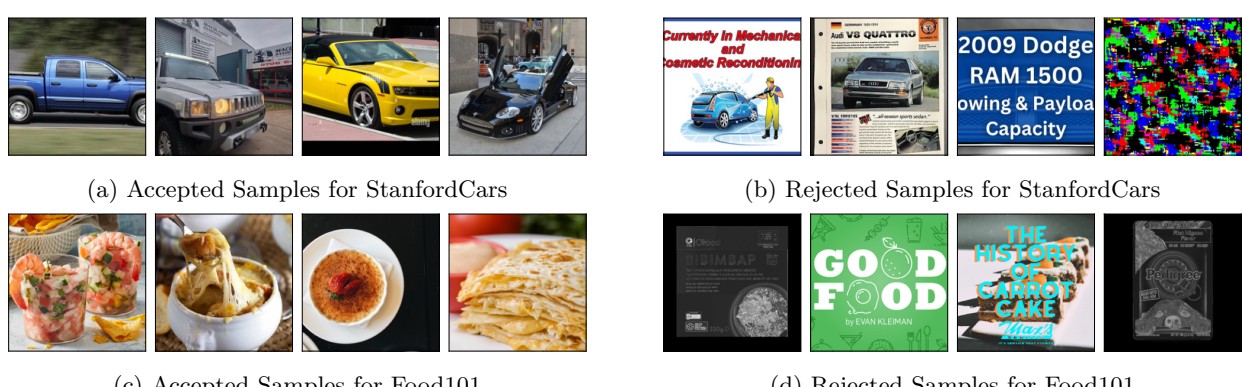

(a) Accepted Samples for StanfordCars

(b) Rejected Samples for StanfordCars

(c) Accepted Samples for Food101

(d) Rejected Samples for Food101

Figure L.16: Random samples from each retrieved dataset based on the refinement procedure described in section 4.3

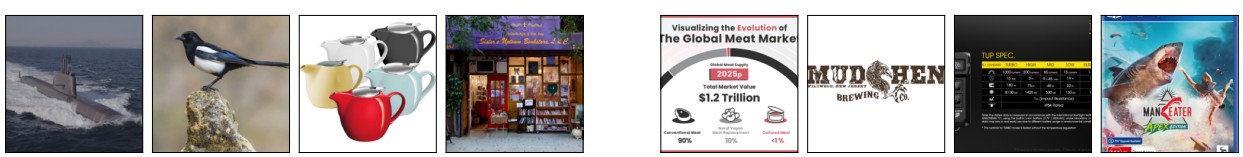

(a) Accepted Samples for ImageNet

(b) Rejected Samples for ImageNet

Figure L.17: Random samples from each retrieved dataset based on the refinement procedure described in section 4.3

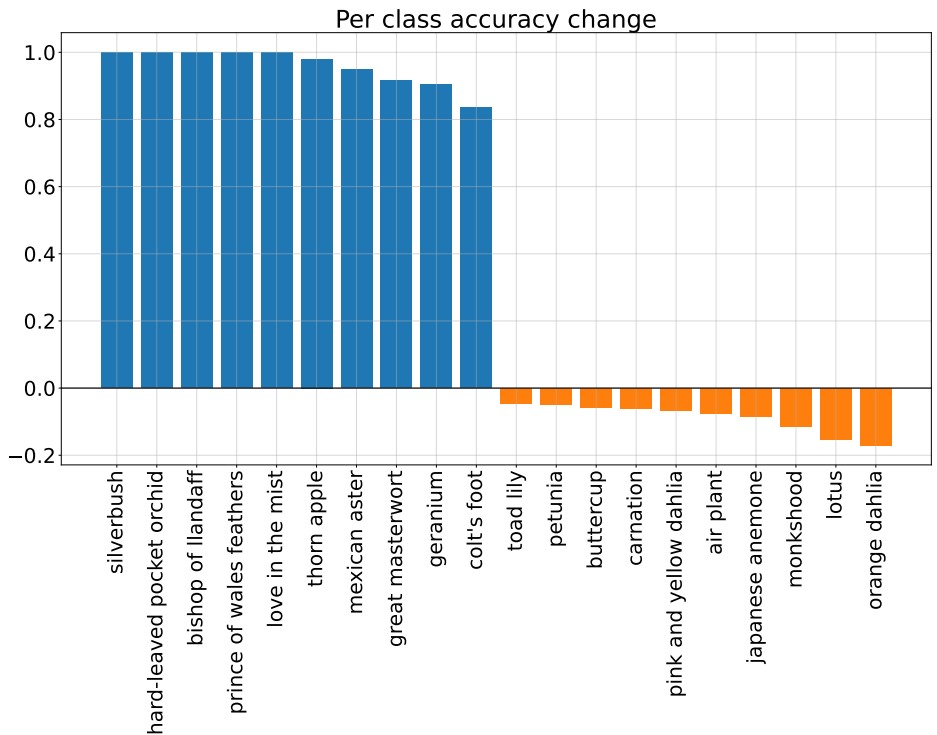

(a) Flowers accuracy change for top-10 and bottom-10 classes

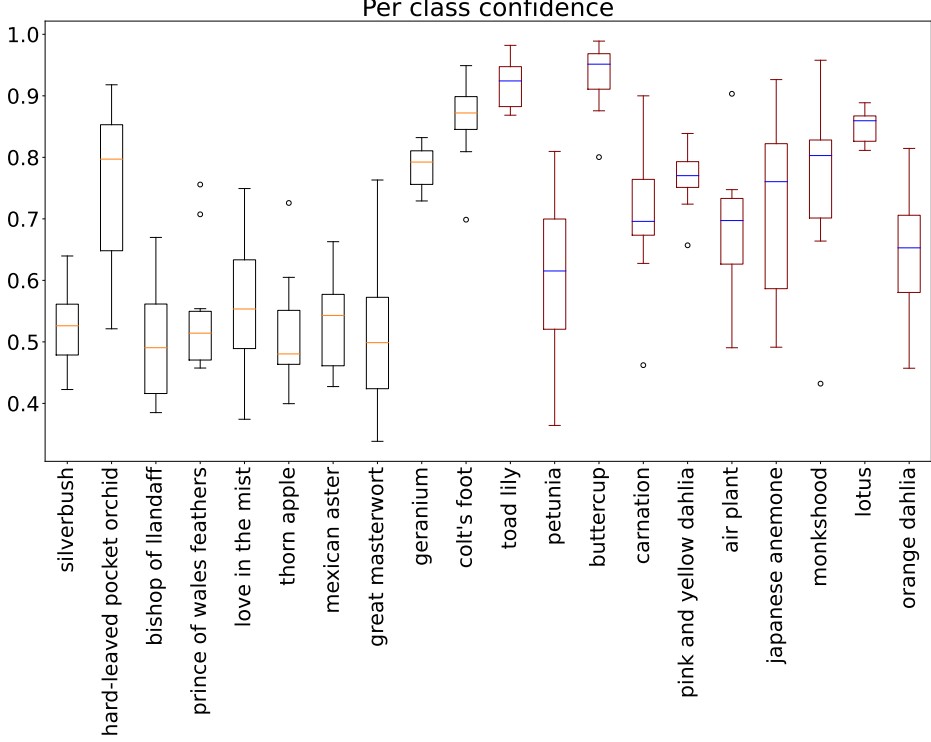

(b) Flowers entropy boxplot for top-10 and bottom-10 classes

Figure M.18: Accuracy improvements and the corresponding confidence values for each class in Flowers.

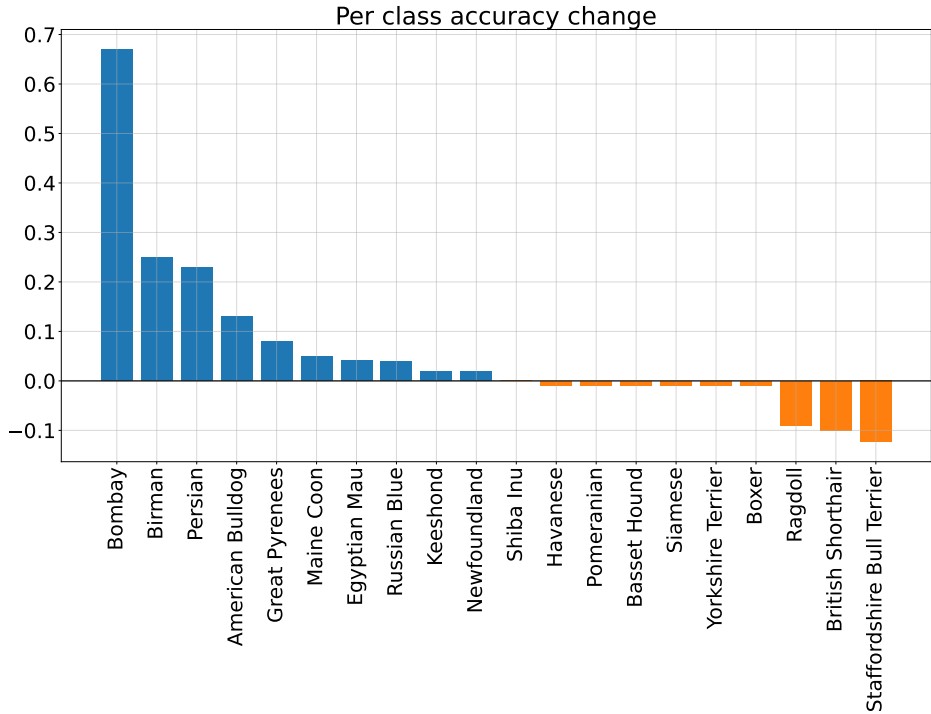

(a) Pets accuracy change for top-10 and bottom-10 classes

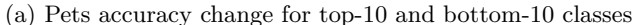

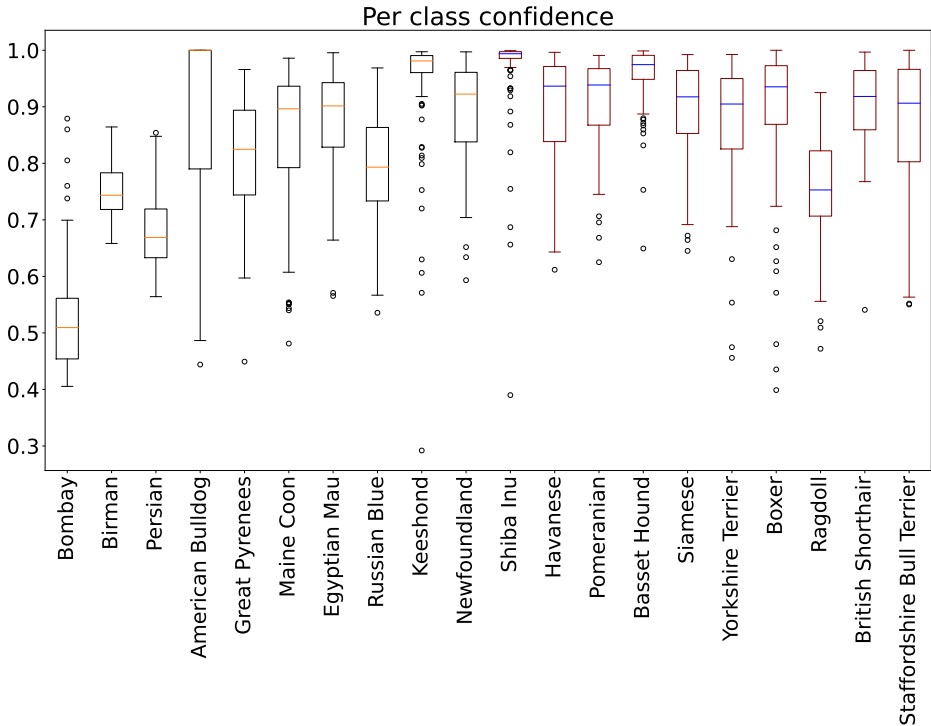

(b) Pets entropy boxplot for top-10 and bottom-10 classes

Figure M.19: Accuracy improvements and the corresponding confidence values for each class in Pets.

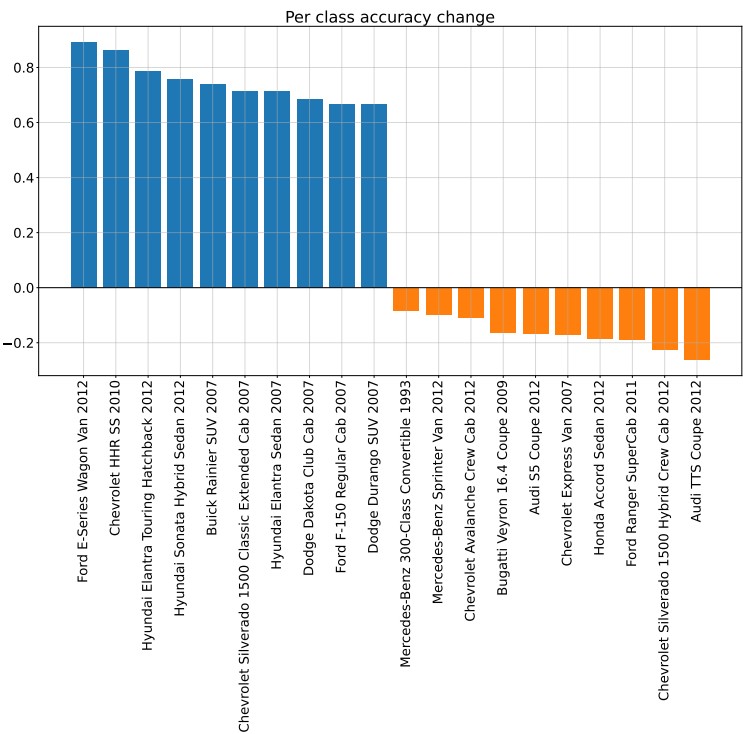

(a) Cars accuracy change for top-10 and bottom-10 classes

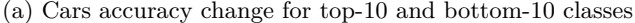
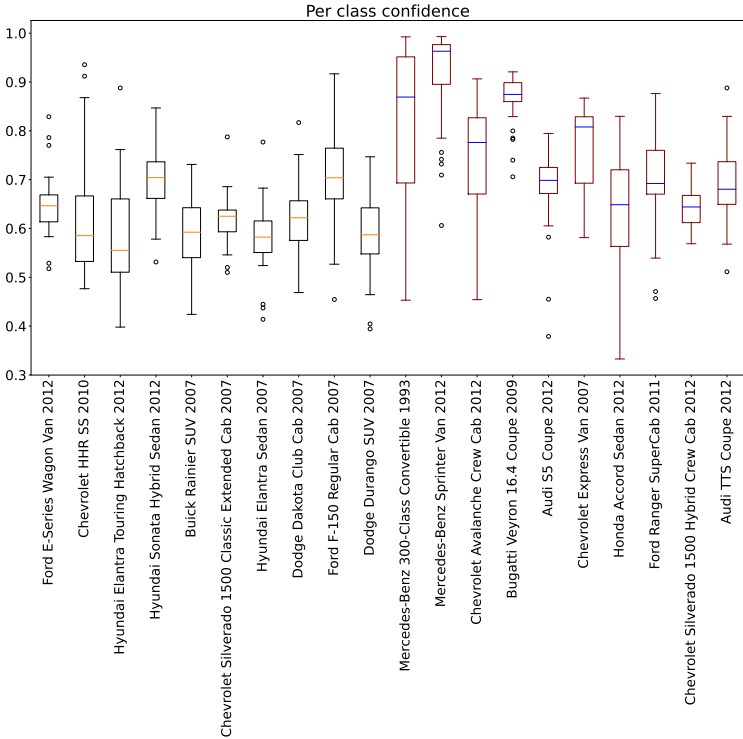

(b) Cars entropy boxplot for top-10 and bottom-10 classes

Figure M.20: Accuracy improvements and the corresponding confidence values for each class in Cars.

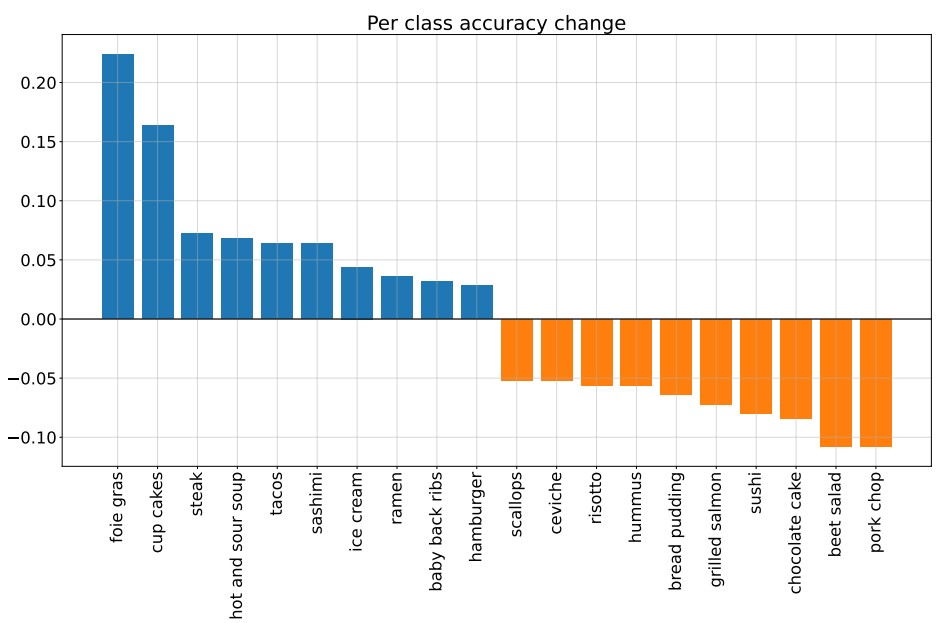

(a) Food101 accuracy change for top-10 and bottom-10 classes

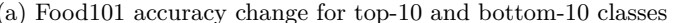

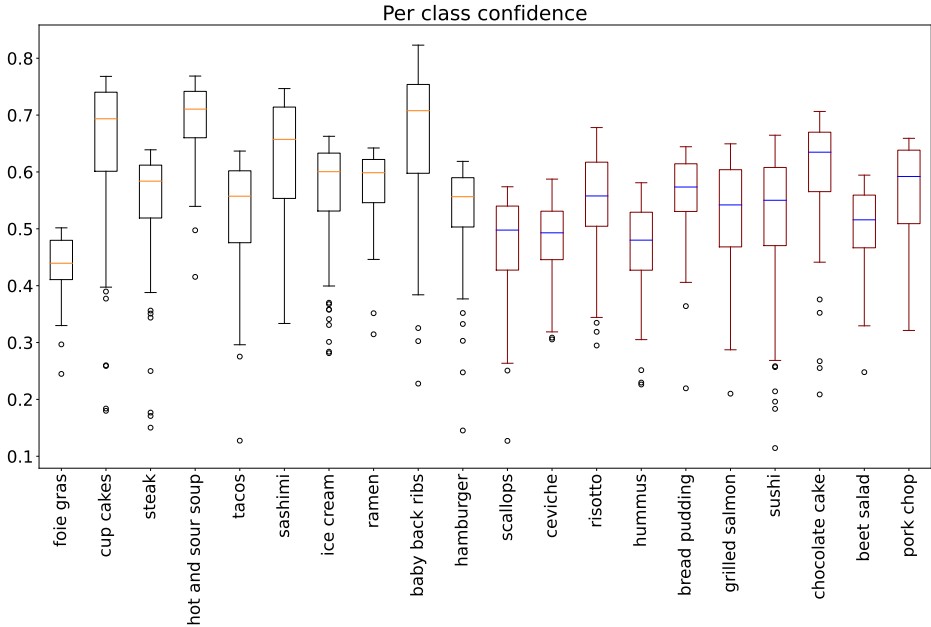

(b) Food101 entropy boxplot for top-10 and bottom-10 classes

Figure M.21: Accuracy improvements and the corresponding confidence values for each class in Food101.

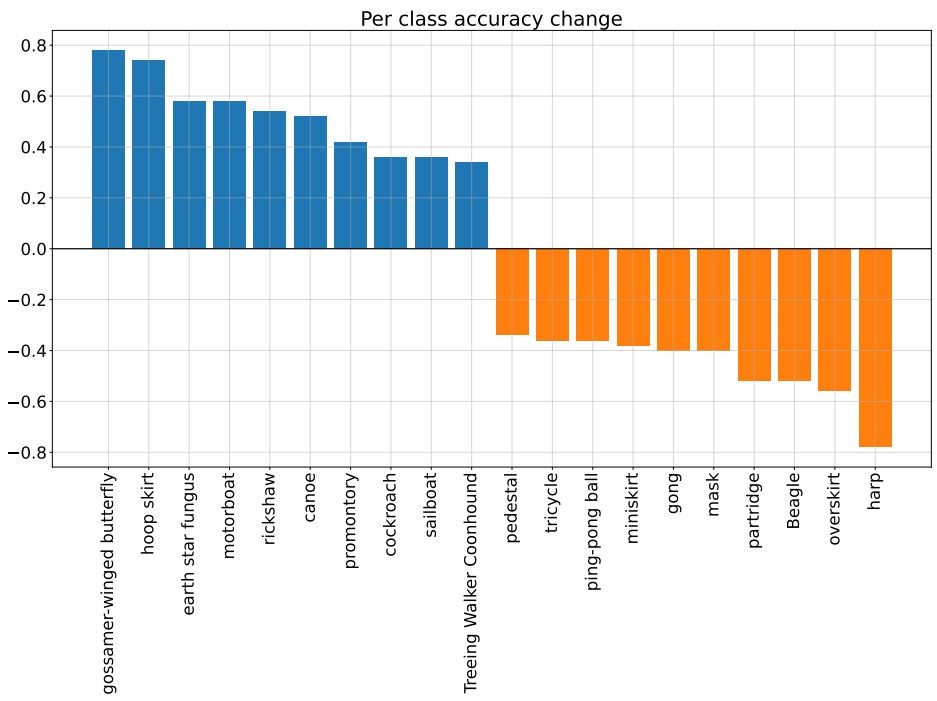

(a) ImageNet accuracy changes for top-10 and bottom-10 classes

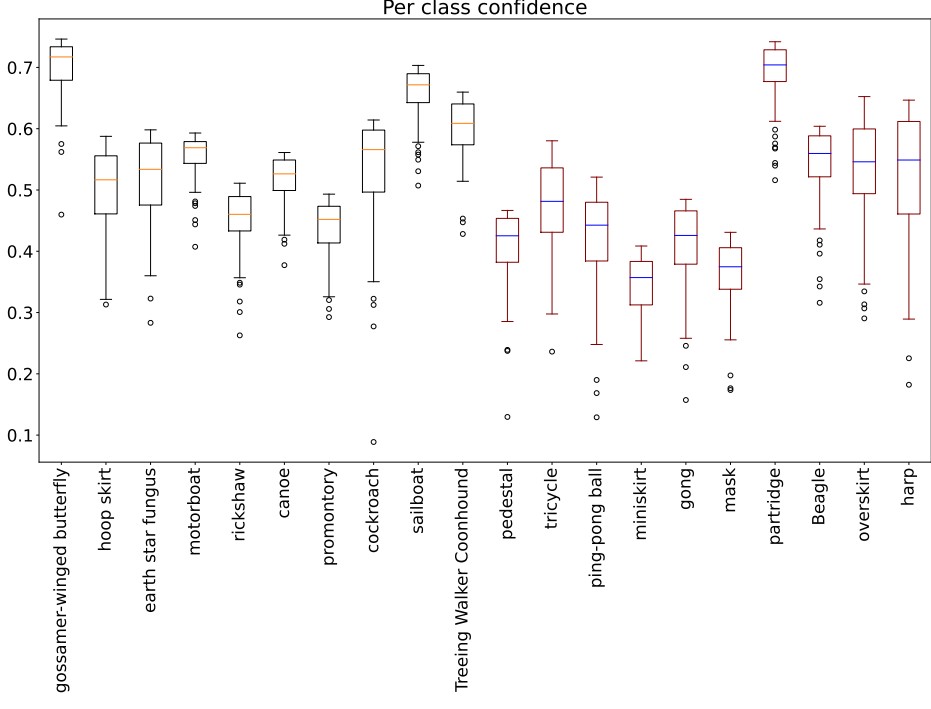

(b) ImageNet confidence boxplot for top-10 and bottom-10 classes

Figure M.22: Accuracy improvements and the corresponding confidence values for each class in ImageNet.

