# OpenReview forum: "Leveraging Uncertainty of Pre-trained Models for Fine-Tuning with Search Engine Retrieval"
_TMLR — Rejected by TMLR_

### Review · Reviewer_ST2s · 2025-03-25

**Summary Of Contributions:**

The paper proposes a method to improve the classification performance of CLIP-like image-text pre-trained models. Specifically, it retrieves images from the web search for classes that the model is uncertain. The retrieved images are then filtered using clustering based on the von Mises-Fisher (vMF) distribution. Finally, these refined images are used to fine-tune the model. The proposed method identifies the uncertain classes by the probability entropy of each unlabeled input image. Three search methods – class names, descriptions, and captioning – are compared, with experimental results demonstrating the simple class name-based approach shows the best performance.

**Audience:**

Yes

**Broader Impact Concerns:**

There is no broader impact statement in the manuscript. Although the research methodology and objectives do not appear to pose significant ethical issues, given that the method relies on web-based image search and download, a discussion on potential ethical implications is needed. For example, images downloaded from the web may inadvertently include copyrighted or otherwise problematic content regardless of the search query.

**Claims And Evidence:**

No

**Requested Changes:**

Please see the weaknesses above.

**<Overall quality>**

* (minor issue) The paper should be carefully polished. There are spelling errors, incorrect capitalizations, inconsistent terminology, etc.
* (minor issue) Section indices in Figure 1 should be “4” instead of “3”.
* (minor issue) Equation (1) seems to be missing a scaling factor of CLIP.
* Overall, the equations need to be refined, and notations should be clarified. For example, in Equation (5), $p, p^{cls}, p_{pre}, p_{\theta}$ are all using “$p$”, but some are actual probabilities and some are not. Also, it is difficult to understand the multiplication-and-sum of “$q_{se}$”, where $p$ is an indicator (0 or 1) and $p^{cls}$ is “conditioned” on label and the dataset X. This “$q_{se}$” is first used in Equation (3) but discussed later in (5).

**<Missing related works>**
* Include a detailed comparison with competitors, especially IE.
* CLIP fine-tuning studies, such as CoOp[1], CoCoOp[2], and TaskRes[3], are missing.
  * [1] Learning to Prompt for Vision-Language Models
  * [2] Conditional Prompt Learning for Vision-Language Models
  * [3] Task Residual for Tuning Vision-Language Models
* Zero-shot image classification studies, such as CuPL[4] and LaBo[5], are missing.
  * [4] What does a Platypus look like? Generating Customized Prompts for Zero-shot Image Classification
  * [5] Language in a Bottle: Language Model Guided Concept Bottlenecks for Interpretable Image Classification

**Strengths And Weaknesses:**

**<Strengths>**
* The proposed web-based retrieval method has potential to be used in various zero-shot image classification tasks.
* The authors conducted an extensive set of experiments (especially presented in Appendix), including ablation studies and tests on different model variants.

**<Weaknesses>**
* The term “uncertainty” is somewhat misleading. The paper appears to conflate the concepts of “out-of-distribution”, “prediction uncertainty”, and “confidence”. Although the paper claims that the method’s uncertainty measurement indicates “label uncertainty”, the entropy-based approach seems to evaluate the level of OOD for input images. Please clarify: (1) what constitutes “uncertain” cases, (2) why the entropy-based approach captures label uncertainty, and (3) how uncertainty, OOD, and web-based search are connected.
* In the main result (Table 2), I am concerned that the improved accuracy may primarily due to the increased amount of training data. Please ensure that all comparisons are conducted with the same amount of training data.
* The reason why web-based images are more beneficial than static images is unclear and not convincing (Table 5). Why do experiments using LAION-5B retrieved data not include LP and LoRA results? What is the key difference between the LAION dataset and web data that makes the performance gap?
* The relationship between “instance-wise” uncertainty and “class-wise” uncertainty is somewhat unclear and loosely connected. In addition, in Section 4.1, why using Top K predictions instead of Bottom K? How are the results from the K x |D| selected labels aggregated?
* The Shannon entropy threshold seems to be used the same for all datasets – but the metric depends on the number of classes in each dataset. Using the same threshold for all datasets would hinder performance improvement and seems quite heuristic.
* The authors tested TIP-Adapter and achieved promising results in the few-shot setup – in fact, the results (Fig. H.14) are somewhat better than those in the main table. This suggests that the training method itself should be examined more carefully. Also, it would be helpful to investigate whether the original dataset (instead of using the retrieved dataset) could also show similar performance for TIP-Adapter. It would clearly demonstrate the contribution of the retrieved dataset.
* Several details are missing, and design choices are not clearly supported. For example, how are the “top n images” selected from the web? What is the importance metric? What is the ‘instance-level captioning’s limitation that necessitates retrieving only the top 5 images to the given query? How is the small set of trainable parameters, theta, established?
* How do we ensure the quality of web-based images? Can we just assume that clustered (similar) images are likely to be correct? Furthermore, since web search result changes over time and region – how does the paper handle this? This is related to reproducibility and long-term applicability.

---

> ### Author Response · Authors · 2025-04-23
> **Author response (Part 1)**
>
> > (1) what constitutes “uncertain” cases, (2) why the entropy-based approach captures label uncertainty, and (3) how uncertainty, OOD, and web-based search are connected.
>
> We use entropy as a proxy for the model's uncertainty in its own predictions. This way, assuming the model has enough knowledge about the general concepts existing in the target dataset, we isolate which classes or instances from the dataset the model is uncertain about. This measure is a commonly used proxy for model uncertainty [a, b, c].
>
> Therefore, we can leave the certain classes or instances while retrieving images only for uncertain classes and fill in the gaps in the model's knowledge of those classes. Given our prediction routing mechanism detailed in Section 4.4, certain predictions are then passed through the base model while uncertain ones are passed through the classification head trained on the retrieved images that fill in the gaps in the model's knowledge of the target datasets.
>
> We do not consider the retrieved images to be out-of-distribution since in that case, we cannot trust the model's entropy to be a sound measure of uncertainty. As mentioned, we assume that the model already has some knowledge of the target dataset and we aim to update its knowledge of the specific classes in the target dataset.
>
> > In the main result (Table 2), I am concerned that the improved accuracy may be primarily due to the increased amount of training data. Please ensure that all comparisons are conducted with the same amount of training data.
>
> The core message of our work is that the gaps in the model's knowledge can be filled in using the retrieved images from the internet. Therefore, by adding more retrieved data using different retrieval methods (such as caption-based and description-based querying detailed in Section 4.2), we fill in the said gap more. In terms of the number of retrieved images per dataset, this value is kept consistent with 100 images per classes for every target dataset. The number of retrieved instances are detailed in Section 5.1.
>
> > The reason why web-based images are more beneficial than static images is unclear and not convincing (Table 5). Why do experiments using LAION-5B retrieved data not include LP and LoRA results? What is the key difference between the LAION dataset and web data that makes the performance gap?
>
> We found the biggest issue when using LAION to be the retrieval mechanism. The internet, or more specifically Google, provides a ranking and retrieval system that, while unknown to the public, works well for the purposes of our method. On the other hand, LAION does not provide a similar high performing retrieval system. Therefore, the user has to design their own retrieval system that would rely on image encodings or image tags that often do not exist within the metadata of the images or the tags are not descriptive enough to be picked up through search. In the meantime, Google is known to extract information about the subject matter of the image from the content of the page, including captions and image titles. This is the main reason that the Google's search engine provides better results compared to their counterparts. Additionally, we note that search engines update information indefinitely while LAION is static and does not get updated.
>
> We provide LoRA results in Table 5 only to show the lack of improvement compared to LP as opposed to providing comparisons to LAION. We show that the model does not need extra information embedded in its weights and instead, only requires updating of its classification head. This is because the performance does not improve when adding LoRA on top of the model's weight matrices and a simple LP improves the performance beyond that of LoRAs. We will separate the table into two for clarity and improve the text.
>
> > The relationship between “instance-wise” uncertainty and “class-wise” uncertainty is somewhat unclear and loosely connected.
>
> Uncertainty for instance refers to the entropy of the predicted distribution for that image. Whereas class-wise uncertainty would refer to the case where images are passed through the model to get the predicted distribution first. Then, we take the top-5 classes predicted by the model for that image and retrieve images for that class.
>
> [a] Mukherjee, Arpan et al. “Uncertainty-Informed Screening for Safer Solvents Used in the Synthesis of Perovskite via Language Models.” (2024).
>
> [b] Wang, Kaizheng et al. “Credal Wrapper of Model Averaging for Uncertainty Estimation on Out-Of-Distribution Detection.” ArXiv abs/2405.15047 (2024): n. pag.
>
> [c] Wang, Kaizheng et al. “CreINNs: Credal-Set Interval Neural Networks for Uncertainty Estimation in Classification Tasks.” Neural networks : the official journal of the International Neural Network Society 185 (2024): 107198.

---

> ### Author Response · Authors · 2025-04-23
> **Author response (Part 2)**
>
> > In addition, in Section 4.1, why using Top K predictions instead of Bottom K? How are the results from the K x |D| selected labels aggregated?
>
> CLIP has a high top-k performance when $k>3$. By choosing the top-k labels, we are targeting the specific classes that the model is confusing for each other. Using bottom-k classes would not solve model's uncertainty in its own prediction.
>
> After refinement, we train the probe on the retrieved images with the label set as the query that was used to retrieve them. This query was obtained from the model's prediction about the uncertain image.
>
> > The Shannon entropy threshold seems to be used the same for all datasets, but the metric depends on the number of classes in each dataset. Using the same threshold for all datasets would hinder performance improvement and seems quite heuristic.
>
> The normalized variant of the Shannon entropy uses a logarithm base equal to the length of the probability vector that represents the predicted distribution from the model. This way, the value of the entropy is always bounded between 0 and 1. Therefore, thresholding works across datasets regardless of the number of classes present in the target dataset.
>
> > The authors tested TIP-Adapter and achieved promising results in the few-shot setup – in fact, the results (Fig. H.14) are somewhat better than those in the main table. This suggests that the training method itself should be examined more carefully. Also, it would be helpful to investigate whether the original dataset (instead of using the retrieved dataset) could also show similar performance for TIP-Adapter. It would clearly demonstrate the contribution of the retrieved dataset.
>
> Our goal is to show the potential of retrieved datasets from the internet using search engines. We use the widely used linear probe to eliminate any shortcomings from more specifically designed classifiers on top of CLIP. Table 2 shows that our approach allows for performance improvements using linear probing and all the tests are performed with a similar setup to allow for reliable conclusions to be drawn with clarity. Adding another method would be redundant.
>
> As shown in Fig. H.14, TIP-adapter does improve performance which is consistent with the TIP-adapter paper, where this method exceeds the performance of LP. Therefore, our existing experiments in Fig. H.14 show that our approach can be further improved with more complex classification heads while the main experiments use a straightforward method to allow for clarity of experimentation and easier analysis of the main proposal which has to do with retrieval mechanisms.
>
> > Several details are missing, and design choices are not clearly supported. For example, how are the “top n images” selected from the web? What is the importance metric?
>
> Top-n images are provided by the search engine. Every search engine provides results after passing them through a ranking system (that is unknown to the user). We take the top-n images from this ranked list based on what the search engine provides as the most related images to our query.
>
> > What is the ‘instance-level captioning’s limitation that necessitates retrieving only the top 5 images to the given query?
>
> Due to the relatively large number of retrieved images, there has to be a cap on the number of images retrieved per uncertain instance. We chose 5 to limit the size of the retrieved dataset. Note that 5 images are retrieved for each top-k class predicted by the model for the uncertain instance. This value can be adjusted for datasets that have more or fewer related images available on the internet.
>
> > How is the small set of trainable parameters, theta, established?
>
> We use linear probing for this purpose following ELEVATER [d]. We freeze all layers except for the last layer while this layer is initialized with the embeddings from the text encoder of CLIP. This way, the classifier starts with the model's pretrained understanding of each class and with each batch, this pretrained knowledge gets updated using the retrieved dataset.
>
> [d] Li, Chunyuan, et al. "ELEVATER: A Benchmark and Toolkit for Evaluating Language-Augmented Visual Models." Advances in Neural Information Processing Systems, edited by S. Koyejo et al., vol. 35, Curran Associates, Inc., 2022, pp. 9287–9301.

---

> > ### Author Response · Authors · 2025-04-23
> > **Author response (Part 3)**
> >
> > > How do we ensure the quality of web-based images? Can we just assume that clustered (similar) images are likely to be correct?
> >
> > Yes, that is the core concept in our refinement mechanism, where after forming distributions over the clustered images, retrieved images too far from the predicted clusters are rejected.
> >
> > > Furthermore, since web search result changes over time and region – how does the paper handle this? This is related to reproducibility and long-term applicability.
> >
> > This is a matter of continual learning which falls outside the scope of the paper. Our goal was using uncertainty for retrieval from the web and propose different mechanisms of retrieval. To keep updating the knowledge of the model of newer available information, we consider proposing a continually updating dataset and probing mechanism that is constantly refined and updated with newer images as our future work.
> >
> > > The paper should be carefully polished
> >
> > We will modify the paper accordingly and improve the writing.
> >
> > > Include a detailed comparison with competitors, especially IE.
> >
> > We provide comparisons to IE in Table 4. Given that IE uses self-supervised training to train another model only to process their retrieved dataset, deeper analysis is not straightforward. This is while we only use a pretrained CLIP, freeze the model except for the updated classification head (while keeping the original weights as a fall-back) and we use CLIP's own feature space for refinement as opposed to training a new model from scratch with a different architecture.
> >
> > > CLIP fine-tuning studies, such as CoOp[1], CoCoOp[2], and TaskRes[3], are missing.
> > [1] Learning to Prompt for Vision-Language Models
> > [2] Conditional Prompt Learning for Vision-Language Models
> > [3] Task Residual for Tuning Vision-Language Models
> >
> > and
> >
> > > Zero-shot image classification studies, such as CuPL[4] and LaBo[5], are missing.
> > [4] What does a Platypus look like? Generating Customized Prompts for Zero-shot Image Classification
> > [5] Language in a Bottle: Language Model Guided Concept Bottlenecks for Interpretable Image Classification
> >
> > Our approach is focused on retrieval mechanisms for updating model's knowledge as opposed to an approach focused on model fine-tuning or improving general classification performance. To this end, we've compared our approach to Internet Explorer, SuS-X, and RECO which are directly related to our work, having the same goal of retrieval and update. The methods mentioned above are alternatives to linear probing as opposed to being alternatives for the retrieval and update process, the latter being the focus of our proposed approach. Therefore, exploring these methods as comparison would be unnecessary and would not provide any insight into our method.
> >
> > > Although the research methodology and objectives do not appear to pose significant ethical issues, given that the method relies on web-based image search and download, a discussion on potential ethical implications is needed
> >
> > We will add a broader impact section discussing the ethical implications, biased data retrieval, IP issues, and environmental impact.

---

> ### Comment · Reviewer_ST2s · 2025-04-30
> **Thank you for the response**
>
> Thank you for your detailed response. They addressed my concerns a lot.
> Below are some comments.
>
> * Thank you for clarifying that the proposed method assumes the target data is already in-domain so that the entropy can measure the label uncertainty. Regardless of the reliability of the assumption, can we interpret that the retrieved images help adjust the output label distribution to fit the target data? I initially thought that the main purpose of the retrieval was somewhat focused on addressing the domain gap between pretrained data and target data, but it does not seem to be the case. Most of my questions were based on this misalignment.
>
>
> * The above question is also related to the paper's expression "gaps in the model's knowledge" - if the gap is not a domain mismatch but a distribution mismatch, then how exactly does the retrieval help reduce the gap? Is this because more samples are added for the selected labels (i.e., the amount of data for each label became more balanced)? Or is this because the diversity of selected labels increased and helped generalization? I'd like to note that these are two different reasons that affect the decision boundary, and it would be nice if the paper could analyze them separately.
>
>
> * It is interesting that in the "Baseline B" experiment (reviewer EdW6), also adding samples to 'certain' classes could lead to worse performance (Pets, Flowers). Maybe, equally increasing the amount or diversity of all classes would hurt the performance. From my perspective, this is not expected if the retrieval helps adjusting the data distribution - certain ones should be more certain.
>
>
> * In addition to the minor issues and math notations (mentioned above), please consider clarifying that the paper is using "normalized Shannon entropy" to avoid confusion.
>
>
> * (You don't need to perform this experiment; just a suggestion) It would be interesting to see whether the retrieval on LAION using CLIP-based image embedding similarity (i.e., no text-based queries) can also help improve the performance. I am somewhat concerned about the heavy reliance on Google search engine's relevance ranking, so it would be nice to show that an alternative approach also works.

---

> > ### Author Response · Authors · 2025-05-04
> >
> > We truly appreciate your ongoing feedback.
> >
> > > can we interpret that the retrieved images help adjust the output label distribution to fit the target data?
> >
> > Yes, since only the classifier is trained as opposed to the model, we don't expect the knowledge embedded in model weights for the dominant patterns of each class to change. Instead, we expect the model to use its already existing knowledge in the pretrained weights with a modified classifier that is trained to help with uncertain classes.
> >
> > > if the gap is not a domain mismatch but a distribution mismatch, then how exactly does the retrieval help reduce the gap?
> >
> > By retrieving based on class name using search engines, we realign the model's classifier to correctly predict the classes it is uncertain about. This gap is due to the model not having seen images closely similar to those of the target dataset despite its existing knowledge about that class. Therefore, due to the diversity of the retrieved images, as you mentioned, generalization performance to the target dataset is improved. Figure 6 in Section 7 shows that while image features from the target dataset form small clusters scattered over the feature space (due to how the target dataset is curated), the retrieved images fill in the gaps in this feature space and make the transition between classes smoother. Therefore, the retrieval process allows for better boundaries to form due to the diversity in the retrieved images for each label.
> >
> > > It is interesting that in the "Baseline B" experiment (reviewer EdW6), also adding samples to 'certain' classes could lead to worse performance (Pets, Flowers). Maybe, equally increasing the amount or diversity of all classes would hurt the performance. From my perspective, this is not expected if the retrieval helps adjusting the data distribution - certain ones should be more certain.
> >
> > In the Baseline B experiment, we are not using our uncertainty-based prediction process described in Section 4.4. The process of training on classes that the model is certain about can lead to overconfident predictions that, on average, have less accuracy than the pretrained model itself. We assume that if the model is already certain about a class, then during pretraining, it has already seen enough images close to that class and retrieving a potentially smaller set of images from the web (compared to the large size of pretraining data) will not have a positive effect. Therefore, in general, more certain predictions does not necessarily correlate with more accurate ones.
> >
> > > In addition to the minor issues and math notations (mentioned above), please consider clarifying that the paper is using "normalized Shannon entropy" to avoid confusion.
> >
> > We will revise accordingly. Thank you.
> >
> > > (You don't need to perform this experiment; just a suggestion) It would be interesting to see whether the retrieval on LAION using CLIP-based image embedding similarity (i.e., no text-based queries) can also help improve the performance. I am somewhat concerned about the heavy reliance on Google search engine's relevance ranking, so it would be nice to show that an alternative approach also works.
> >
> > We believe this is a valid future trajectory and prompts a work that compares the sources of information that retrieval uses. We will mention this in our conclusion section.

---

### Review · Reviewer_sqW9 · 2025-04-05

**Summary Of Contributions:**

This paper proposes a method to enhance pre-trained models (specifically CLIP) by retrieving additional data from web search engines based on model uncertainty. The approach first identifies instances where CLIP is uncertain about its predictions, then generates search queries for these cases, retrieves relevant images from the web, refines them using a geometry-aware approach, and finally fine-tunes a small model on these refined images. The authors demonstrate significant performance improvements on several datasets, particularly StanfordCars and Flowers (15+ percentage points increase in accuracy), while requiring two orders of magnitude less data compared to existing methods. They also analyze the impact of different query construction strategies, refinement techniques, and the relationship between model uncertainty and performance improvement.

**Audience:**

Yes

**Broader Impact Concerns:**

The paper doesn't include a Broader Impact Statement, which is a significant omission. Several ethical implications should be addressed:

The method relies on web search engines, which may propagate or amplify biases present in web data. This is particularly concerning for classification tasks involving people or cultural artifacts.
There are potential copyright and intellectual property issues when retrieving and using images from the web for model training, even if at a small scale.
The approach might lead to data leakage issues beyond what's discussed in Section 6. While the authors check for direct image matches, semantic leakage (where conceptually similar images appear in both test sets and web results) is not addressed.
Reliance on third-party search engines raises questions about reproducibility and stability of results over time, as search algorithms and indexes continuously change.
The environmental impact of retrieving and processing additional images from the web should be considered, especially if this approach were to be deployed at scale.

**Claims And Evidence:**

Yes

**Requested Changes:**

Critical Changes:
Strengthen the motivation for the approach in Section 1. Explain why this method would remain valuable even as pre-trained models grow larger. Perhaps discuss specific domains or applications where even the largest models might still benefit from web retrieval.
Provide a more rigorous theoretical justification for why retrieving images for uncertain classes specifically leads to better performance. This should be added to Section 4.1 where uncertainty is discussed.
Expand the experimental comparisons in Section 5.4 to include more baselines, particularly methods that use alternative fine-tuning strategies or prompt engineering approaches that don't require additional data.
Restructure the paper to improve flow and organization. The question of "what tasks are expected to benefit?" (currently in Section 5.2) should be moved to the methodology section as it's crucial for understanding when to apply this approach.

Suggested Improvements:
Include a more detailed analysis of the computational costs associated with the approach, including search engine API costs, refinement time, and fine-tuning overhead.
Provide more insights into why the approach works well for some datasets (Cars, Flowers) but not others (Food, ImageNet) beyond the brief discussion in Section 5.5.
Explore how the approach might complement other methods for enhancing pre-trained models, rather than presenting it as a standalone solution.
Add more discussion about potential failure cases and limitations, particularly for specialized domains where web search might not yield relevant results.
Consider adding ablation studies that isolate the contribution of each component (uncertainty-based selection, query construction, refinement) to the overall performance improvement.

**Strengths And Weaknesses:**

Strengths:
The idea of leveraging search engines to automatically retrieve task-specific data is interesting and practical.
The approach doesn't require any additional labeled data or manual intervention.
The improvements shown on some datasets (Cars and Flowers) are substantial.
The proposed refinement technique (using von Mises-Fisher distributions) is mathematically sound.
The paper includes detailed analyses on various aspects of the approach, including uncertainty threshold effects, refinement methods, and the number of images per query.

Weaknesses:
The motivation for this approach seems insufficient in the era of increasingly larger pre-trained models. As pre-trained models continue to grow in size and training data diversity, one can expect them to eventually cover the datasets used for validation in this paper, potentially making this approach less necessary (not adequately addressed in Section 1).
The paper lacks a strong theoretical foundation. While the method works empirically, there's little theoretical explanation for why searching for uncertain classes specifically leads to better performance. The von Mises-Fisher distribution is used for refinement (Section 4.3), but the paper doesn't rigorously justify this choice beyond noting that embeddings lie on a hypersphere.
The experimental comparisons are somewhat limited. While the authors compare with a few recent methods like Internet Explorer, SuS-X, and RECO (Table 4), the comparison doesn't include other potential approaches like prompt engineering or other fine-tuning methods that could achieve similar improvements.
The paper's organization is somewhat confusing. For example, Section 5.2 asks "What tasks are expected to benefit?" but this critical question about the method's applicability should be addressed earlier, perhaps in the methodology section.
The performance improvements are inconsistent across datasets. While significant gains are shown for Cars and Flowers, the improvements on Food and ImageNet are minimal (Table 2). This raises questions about the generalizability of the approach.
There's a lack of discussion about computational costs. While the method requires fewer images, the process of retrieving images from search engines, refining them, and fine-tuning models adds computational overhead that isn't thoroughly analyzed.

---

> ### Author Response · Authors · 2025-04-23
> **Author response (Part 1)**
>
> > Strengthen the motivation for the approach in Section 1. Explain why this method would remain valuable even as pre-trained models grow larger. Perhaps discuss specific domains or applications where even the largest models might still benefit from web retrieval
>
> We have shown in Table F.7 that larger models like ViT-g-14 still consistently benefit from our proposed method even though this model is larger than ViT-B-32 by over a magnitude and is trained on a significantly larger dataset.
>
> Regardless of how large a pre-trained model gets and how much information it holds, newer information will come out. Additionally, similar information to what the model has already seen can be presented differently to the model causing uncertainty in its predictions, leading to wrong predictions due to the concept of productivity in compositionality of data (infinite combinations of data is possible). Therefore, new information will be available on the internet and either this new information has to be distilled into the model by further training or a mechanism has to exist to update the information based on what the model knows and doesn't know. We propose an uncertainty-based mechanism that addresses this issue by retrieving information that the model is uncertain of in order to update the existing information within the weights of the model.
>
> > Provide a more rigorous theoretical justification for why retrieving images for uncertain classes specifically leads to better performance. This should be added to Section 4.1 where uncertainty is discussed
>
> We use uncertainty as a measure delivered by the model about what it knows and what it doesn't know about the datasets passed through the model. This approach is straightforward and widely used as a measure for uncertainty. Retrieving images for classes that the model is certain about would have no practical advantage since the model is already capable of predicting accurately and our goal is to fill in the gaps in the model's knowledge as opposed to performing model editing. Due to the straightforwardness of our proposed method, we don't believe a theoretical foundation is necessary.
>
> > The von Mises-Fisher distribution is used for refinement (Section 4.3), but the paper doesn't rigorously justify this choice beyond noting that embeddings lie on a hypersphere
>
> CLIP embeddings lie on the unit hypersphere. Therefore, applying clustering methods that do not operate on the manifold would be inappropriate since methods such as KNN are designed with the assumption that data lies in a flat space. We use the vMF distribution which operates on unit spheres and allows for parametric clustering [a]. As far as we are aware, [a] is one of the only methods that proposes a parametric clustering approach with a principled on-manifold distribution-based approach. Other approaches (such as [b]), while capable of performing clustering on-manifold, are not specifically designed for the unit hypersphere.
>
> [a] Banerjee, Arindam, et al. Clustering on the Unit Hypersphere Using von Mises-Fisher Distributions. Journal of Machine Learning Research.
>
> [b] Nielsen, Frank. "Hierarchical Clustering." 2016, pp. 195–211.

---

> ### Author Response · Authors · 2025-04-23
> **Author response (Part 2)**
>
> > the comparison doesn't include other potential approaches like prompt engineering or other fine-tuning methods that could achieve similar improvements
>
> and
>
> > Expand the experimental comparisons in Section 5.4 to include more baselines
>
> The mentioned methods modify the model to the extent that catastrophic forgetting becomes an issue. As a reminder, our goal is to improve the performance of the model by updating the information about uncertain classes. Our approach does not modify the model so far as to remove generalizability of the model to other tasks. Both fine-tuning and some prompt engineering methods would alter the model to the extent that performance on other tasks would degrade due to the focus on the new target task.
>
> ---
>
> > The paper's organization is somewhat confusing. For example, Section 5.2 asks "What tasks are expected to benefit?" but this critical question about the method's applicability should be addressed earlier, perhaps in the methodology section.
>
> We will modify the paper's structure accordingly.
>
> ---
>
> > The performance improvements are inconsistent across datasets. While significant gains are shown for Cars and Flowers, the improvements on Food and ImageNet are minimal (Table 2).
>
> We obtain more than 3\% improvement on ImageNet1k which is significant. On Food101, as mentioned in Section 5.2 and Table 1:
>
> > the span of labels in Food forms the narrowest cone in the CLIP feature space among all the studied datasets suggesting either low specificity or insufficient training data per category
>
> This shows that we identified that Food101 will not get significant improvements before retrieval due to low specificity of the dataset. This makes it difficult to find specific images online that will match the distribution of Food101. Meanwhile, based on Table 2, we obtain consistent improvements when using all retrieved datasets for all target datasets. We will explain further in the paper how Food101 is disimilar to other datasets in this manner.
>
> ---
>
> > There's a lack of discussion about computational costs. While the method requires fewer images, the process of retrieving images from search engines, refining them, and fine-tuning models adds computational overhead that isn't thoroughly analyzed.
>
> Since we use a linear probe on top of the model (a single fully-connected layer after the final layer of the model used as the classification head of CLIP itself), the cost of training and inference are the same as CLIP. Refinement is done using the scikit-learn implementation of vMF clustering method which takes little time to perform. The only process that takes noticeable amount of time to complete is retrieval. Based on our experiments, when using the readily available libraries for retrieval from google, we needed 3 days for ImageNet for class-based, caption-based, and description-based retrieved datasets. This is while class-based dataset took less than 6 hours to complete and the majority of overhead was due to the instance-based retrieval for caption-based and description-based methods. We will add a table detailing the expected amount of time for each dataset to the appendix.
>
> ---
>
> > Explore how the approach might complement other methods for enhancing pre-trained models, rather than presenting it as a standalone solution
>
> As mentioned, our goal is to update the model's classifier with the new data that the model is uncertain about. Incorporating our appraoch into pretraining would go against the core of our proposed approach.
>
> ---
>
> > Add more discussion about potential failure cases and limitations, particularly for specialized domains where web search might not yield relevant results
>
> We will add a discussion on this matter to the conclusion section.
>
> ---
>
> > Consider adding ablation studies that isolate the contribution of each component (uncertainty-based selection, query construction, refinement) to the overall performance improvement.
>
> This ablation already exists in the paper. Table 2 shows the effect of adding and removing refinement. Table 2 also shows the effect of adding each retrieved dataset one by one to show the improvement from each retrieval mechanism (query construction). Without uncertainty-based selection, our approach would obtain the same results since our fallback mechanism detailed in Section 4.4 routes the inputs through the original model for certain predictions.
>
> ---
>
> > The paper doesn't include a Broader Impact Statement, which is a significant omission. Several ethical implications should be addressed:
>
> We will add this section and discuss the ethical implications, biased data retrieval, IP issues, and environmental impact. We're not sure of what semantic leakage means here. Technically, any training dataset is semantically related to the test set and therefore any data that contains information remotely related to the test classes will have "semantic leakage".

---

> > ### Comment · Reviewer_sqW9 · 2025-05-25
> > **Response to Authors' Rebuttal**
> >
> > Thank you for your detailed responses to my review. After careful consideration, I must maintain my original assessment that this paper lacks sufficient theoretical justification for a TMLR publication.
> >
> > My primary concern about the theoretical foundation remains inadequately addressed. While you state that your uncertainty-based approach is "straightforward and widely used" and that you "don't believe a theoretical foundation is necessary," I respectfully disagree. For a journal-level publication in TMLR, it is essential to provide rigorous explanations for why the proposed methodology works beyond empirical observations.
> >
> > Specifically:
> >
> > 1. The paper still lacks a theoretical analysis of why retrieving images for uncertain classes leads to better performance compared to alternative strategies. Simply stating that adding data for certain classes has "no practical advantage" is insufficient without formal justification.
> >
> > 2. Regarding the von Mises-Fisher distribution, while you explained that CLIP embeddings lie on a hypersphere, the paper should provide a more thorough analysis of why this particular distribution is optimal for your refinement process compared to other manifold-aware clustering approaches.
> >
> > 3. The relationship between uncertainty, retrieval strategy, and performance gains remains largely empirical rather than theoretically grounded. A deeper analysis of this relationship would significantly strengthen the paper.
> >
> > While your method shows promising empirical results, TMLR publications typically require stronger theoretical underpinnings to explain why methods work rather than just demonstrating that they do. Therefore, I maintain my assessment that in its current form, the paper lacks necessary explanations for the proposed methodology and is not yet suitable for TMLR.

---

> ### Author Response · Authors · 2025-06-13
>
> We appreciate the reviewer's emphasis on theoretical rigor and have addressed these concerns by providing a formal information-theoretic justification for our uncertainty-based retrieval approach in Appendix A. References to this appendix section have also been added to section 4.1 and section 4.4 in the revised text.
>
> **Why uncertainty-based retrieval works**:
> Our theoretical analysis establishes that uncertainty-based retrieval is not merely heuristic but mathematically optimal. We prove in Proposition 1 that for any two instances $x_1$ and $x_2$ where $\mathcal{H}(Y|x_1) > \mathcal{H}(Y|x_2)$, the expected information gain from retrieved data satisfies $\mathbb{E}[\text{IG}(Y; X_{\text{retrieved}} | x_1)] \geq \mathbb{E}[\text{IG}(Y; X_{\text{retrieved}} | x_2)]$. This monotonicity property directly follows from the fundamental relationship between initial entropy and maximum achievable information gain.
> The key insight is that information gain is upper-bounded by the initial predictive entropy, i.e., $\text{IG}(Y; X_{\text{retrieved}} | x) \leq \mathcal{H}(Y|x)$. Therefore, instances with higher uncertainty have a higher ceiling for potential improvement. Under the reasonable assumption that search engines return comparably relevant results across queries, this upper bound translates to higher expected gains for uncertain instances.
> Corollary 1 extends this to prove that given a fixed retrieval budget B, selecting instances with the highest predictive entropy maximizes the total expected information gain. This provides the formal justification for why retrieving data for certain classes (low $\mathcal{H}(Y|x)$) offers minimal benefit, their low entropy leaves little room for information gain, making such retrieval suboptimal under resource constraints.
>
> **Relationship between uncertainty, retrieval, and performance**:
> Our theoretical framework establishes a clear connection between these components through information theory. The relationship operates as follows:
>
> - Uncertainty Quantification: $\mathcal{H}(Y|x)$ measures the model's uncertainty about instance x
> - Retrieval Optimality: Higher $\mathcal{H}(Y|x)$ instances provide greater expected information gain $\mathbb{E}[\text{IG}(Y; X_{\text{retrieved}} | x)]$
> - Performance Improvement: Information gain directly translates to performance through entropy reduction, as shown in our prediction mechanism (Eq. (8))
>
> The information-theoretic framework thus provides both the mathematical justification for our approach and predictive insight into when the method will be most effective, bridging the gap between theoretical understanding and empirical performance.
>
> **Clustering on a hypersphere**:
> CLIP embeddings, by design, lie on the unit hypersphere due to L2 normalization during contrastive training. For our refinement process, we require a clustering approach that provides: (1) a parametric distribution representation of clusters, and (2) a principled method to quantify membership likelihood for thresholding retrieved instances.
> The von Mises-Fisher (vMF) distribution is the natural generalization of the Gaussian distribution to the unit hypersphere, making it theoretically optimal for our setting. Specifically, vMF distributions maximize entropy subject to constraints on the mean direction and concentration, providing the least biased representation of directional data clusters. This is crucial because our refinement process relies on computing $p(x_r | X)$ to determine whether retrieved images $x_r$ belong to the same distribution as the target dataset $X$.
> The parametric nature of vMF allows us to compute exact likelihoods $p_{μ_i,\kappa_i}(x_r)$ for cluster assignment, rather than relying on ad-hoc distance metrics. The concentration parameter $\kappa_i$ naturally captures cluster tightness, enabling principled thresholding via $\tau_R$ in Equation (6). This probabilistic foundation ensures that our refinement process has a clear statistical interpretation. We reject samples that have low probability under the mixture model representing the target distribution.
> We provide empirical comparisons with alternative approaches in Appendix E, including tangent space k-means and text-based clustering. While these methods can approximate the desired behavior, they require additional assumptions (local flatness for k-means) or lose the geometric structure of the embedding space (text-based methods). The vMF approach directly operates on the native geometry of CLIP embeddings without approximations, making it both theoretically principled and practically effective for our hyperspherical data.

---

> > ### Comment · Reviewer_sqW9 · 2025-06-14
> > **Request for Stronger Theory**
> >
> > While the authors have developed some seemingly interpretable theoretical components, I believe there is room for further reinforcement through enhanced theoretical construction and validation. Therefore, I am revising my evaluation upward to borderline.

---

### Review · Reviewer_EdW6 · 2025-04-08

**Summary Of Contributions:**

The paper proposes a method of finetuning CLIP-like models to a dataset. For an instance that the model is confident, the method lets the model do the classification. for the instances that the model is less confident it searches the web for new training data. It trains a classifier on top of CLIP and it tries to classify the instances. If the classifier is confident the instances are classified, else the original CLIP classification occurs.

**Audience:**

Yes

**Broader Impact Concerns:**

No other concerns

**Claims And Evidence:**

No

**Requested Changes:**

As a feedback:
- I think W3 should be resolved

Critical for Acceptance:
- Resolving W1 and W2 are critical in my opinion for acceptance. W4 is critical for a camera ready but trivial. Note that W1 made me be incapable of fully understanding the claims of this paper and therefore I have to set the "Claims And Evidence" as "No". I am willing to change it after W1 is resolved

**Strengths And Weaknesses:**

Strangths

- S1: The setting is reasonable and useful. It is reasonable to assume internet acces in many realistic senarios and indeed the web is the ultimate dataset.
- S2: the paper have many experiments to showcase

Weaknesses
- W1: In my opinion the most important experiment is missing. The trivial extreme case of this method is to offline build a dataset from web images for every class (same images per class as your method) and finetune CLIP with that. This of course is easy but not interesting, let's call it Baseline B. Your method is selecting less data from the web based on the uncertainty of CLIP. If we don't know whether or not your method is better or worse than B, the argumentation of its usefulness will be missing. I think you should do this experiment (B) and then claim 1) are you better (in Acc) than B? 2) are you faster than B? are you using less data (yes) than B? In general, the question that should be answered is "why should anyone follow your method and not do B?"
- W2: 5.5 "trained on the best retrieved dataset" I don't think you get to choose a method per dataset. If you need that you should propose a principled automated way for that choice. Access to the test set should be out of the question.
- W3: In 4.1 it is not clear to me if the X_uncertain includes test set images. If yes, having to use information from the whole manifold of the test set is a limitation and it should be addressed in the script. If it includes only training images then it should be clear in the script.
- W4: Typos and other mistakes. Indicatively: 1) In introduction line3: Is DINO multimodal? did you mean CLIP? 2) In page 3 Google Align is Chao Jia et. al., not Radford. 3) CLIP is ICML 2021, not ArXiv.

---

> ### Author Response · Authors · 2025-04-23
>
> W1: This depends on the retrieval method. For $\mathbf{D}^{\text{cls}}_{\text{uncertain}}$, including retrieved images for all classes does not change the accuracy. This is because the model is already confident in its predictions for those classes, and our prediction routing mechanism (Eq. 8) defaults to the original model when confidence is high. In the case of instance-based retrieval datasets (caption-based and description-based), the linear probe's performance gets worse due to disregarding uncertainty. This is because the retrieved images based on instances of confident predictions greatly outnumber that of uncertain ones, leading to a training set that does not meaningfully improve accuracy. Therefore, using Scenario B would mean unnecessary retrieval of images that will end up degrading the performance.
>
> W2: The following quote:
> >trained on the best retrieved dataset
>
> Is mentioned in Section 5.3 in an isolated experiment. The goal here is to assess how close the best possible results get to the ideal scenario (training on the train set of the dataset alongside the retrieved images). This helps determine whether the retrieved images truly complement the training data or just duplicate existing information. The results indicate that the retrieved images indeed enhance the performance of the linear probe, even when used alongside the original train split. As for the main method, we mention across the paper, particularly in the introduction, that:
> >We further found if the task is to benefit from our approach, the best margin of improvement is achieved when the class name (provided by the dataset) is used as the search query rather than more complicated alternatives such as descriptions or instance-level queries from image captioning.
>
> Stating that class based retrieval provides the best results while retrieving the fewest number of images. This is while the best results regardless of the number of retrieved images is obtained through the combination of all proposed retrieval methods.
>
> W3: We have tested both train set and test set images for X_uncertain and the rejection rate and the performance do not change.
>
> W4: Thank you, we will modify the paper accordingly.

---

> ### Comment · Reviewer_EdW6 · 2025-04-23
>
> Dear authors,
>
> Thank you for the answers above. I accept the answers for W2 and W4. For the other two, I need to see the following in order to be convinced:
>
> W1: If I wanted to solve a classification problem and I had access to the internet, the first thing that would cross my mind is to use my class names to create a dataset from the web. Then I would finetune a good backbone, let's say CLIP. This is what I call Baseline B (not downloading every class and then use your method that will reject the classifier output). You are proposing something quite more complicated than B. I think you should show an experiment (and include it in the paper) that demonstrates that your method is better than B.
>
> W3: The answer is not clear to me. Is your final method using the manifold of the test set during test time? in other words, in order to classify a single test image, do you need more test images to be present? if yes, that's ok but you should specify it as a limitation

---

> > ### Author Response · Authors · 2025-04-25
> >
> > We sincerely appreciate your continued feedback. We clarify each point further below.
> >
> > W1: The results of Baseline B are shown here compared to our approach from Table 2 and Zero-Shot results.
> > |                | Imagenet | Flowers | Cars  | Pets  | Food101 |
> > |----------------|----------|---------|-------|-------|---------|
> > | Zero-shot      | 68.33    | 71.15   | 64.71 | 89.04 | 88.73   |
> > | Our Approach   | 69.05    | 86.24   | 80.52 | 92.61 | 88.97   |
> > | Baseline B     | 57.93    | 85.79   | 80.45 | 90.79 | 81.13   |
> >
> > The performance on ImageNet and Food101 severely degrades without using our proposed pipeline. As mentioned in the paper,  classes present in Food101 have low specificity, i.e. for each label, the distribution of the images belonging to that class can be broad. This makes it difficult to find images that closely match the distribution of the specific images provided in the dataset. This was known to us before any linear probing experiments due to the results in Table 1, section 5.2. ImageNet, on the other hand, consists of a large number of classes (1000 classes) and without refinement and uncertainty focused retrieval and prediction, the trained probe has to deal with significant levels of noise which degrades its performance. For Pets and Flowers, we observe slight performance degradations which have to do with disregarding the classes that the model is certain about. Performance on Cars, on the other hand, is not affected by this scenario, showing that when the labels are highly specific (e.g. Toyota Corolla 2012 Sedan), the retrieved images are appropriately representative of the dataset. Moreover, due to the nature of this dataset, high quality images are easy to come by on the internet. We will add these results to the paper as a new appendix section with references to this section in the main text.
> >
> > W2: In the main text, we assume access to the test images (not labels) is allowed and use this set for clustering. However, as mentioned, when swapping the test images for train images, the same images are rejected and the same exact performance is achieved. We will clarify this in the paper.

---

### Comment · Action_Editor_Sb6K · 2025-04-23
**Rebuttal**

Dear authors,

Can you please give an indication for response to reviews? Or shall the reviewers submit their final recommendation already?

Thanks,
Your Action Editor

---

### Author Response · Authors · 2025-04-23

We sincerely thank all the reviewers for their thoughtful and constructive feedback. We appreciate the recognition of our method’s practicality and potential, particularly in leveraging web data to augment vision-language models without manual labeling. We are grateful to Reviewer EdW6 for recognizing the practicality of our setting and the breadth of our experimental results. We thank Reviewer sqW9 for highlighting the effectiveness of our approach, the substantial improvements on certain datasets, and the soundness of our refinement method. We also thank Reviewer ST2s for noting the method's applicability to zero-shot classification tasks and the depth of our experimental analysis. We appreciate each reviewer's time and effort in evaluating our work and have carefully addressed their comments in response to each review.

---

### Author Response · Authors · 2025-05-10
**Revised paper**

In the newly revised version of our paper, we have thoroughly addressed the reviewers’ suggestions and incorporated several improvements to strengthen the paper. Building on the valuable feedback provided, we have made the following changes:

- Revised the introduction to include a discussion of larger models while mentioning the target focus of our work on in-domain generalization.
- Improved the clarity and flow of Sections 3 and 4. We also added a temperature scaling parameter to Eq. (1) and revised the writing to better explain the formulation.
- Simplified and clarified the mathematical representation of our method in Eqs. (2), (3), and (5), and unified notation for retrieval and search-engine-related sampling.
- Introduced a new Section 9 on limitations, discussing issues such as specialized domains, out-of-distribution data, search engine variability, ambiguous class names, and retrieval overhead.
- Added a new Section 10 on broader impact, covering topics including copyright and IP concerns, web scraping ethics, retrieval bias, and environmental considerations.
- Added Appendix B presenting an ablation study on uncertainty as a retrieval mechanism, highlighting its importance and analyzing the reason for variability in performance across datasets. References made to this appendix are in section 5.7.
- Added Appendix E detailing retrieval-related technical considerations, such as expected retrieval time and API cost estimates, with references to this appendix in section 5.1.
- Corrected references to Section 4 in Figure 1.
- Carefully reviewed and fixed stylistic inconsistencies throughout the paper, e.g., ensuring consistent lowercase usage for terms like “web” and “internet.”
- Corrected citations of CLIP.

Note: Given the current structure and narrative flow of the paper, we believe moving Section 5.2 before the experiments, and thus before the dataset introduction, would disrupt the coherence of the presentation and potentially confuse readers. For this reason, we have retained Section 5.2 in its original position relative to the rest of the paper.

We hope these revisions address the reviewers' concerns and improve the quality and clarity of our work. We sincerely appreciate the time and effort the reviewers and editors have dedicated to evaluating our submission.

---

### Decision · Action_Editor_Sb6K · 2025-08-06

**Recommendation:** Reject

**Audience:**

Yes

**Audience Explanation:**

The topic is certainly of interest, as data is getting harder to collect and this fine tuning is also becoming more and more expensive.

**Claims And Evidence:**

No

**Claims Explanation:**

The reviewers looked into the paper and reviews are mixed, with two or of three reviews both criticizing lack of clarity and rigor. As these comments can be seen as vague, I went through the paper myself.

Reading the paper, I see that the definition of uncertainty is defined on the basis of normalized Shannon entropy, where families with high entropy are labeled as uncertain. That is, the method that does use bayesian models to quantify uncertainty, and rather relies on a more practical definition, assuming that prediction confidence is the same as uncertainty. This is not necessarily the case, as the confidence of prediction is a function of all classes in the dataset (for instance, as the number of classes increases, the prediction confidence drops while the image remains the same).

I think the authors should do a major revision and change the positioning so that it is not about uncertainty but about Shannon entropy and/or prediction condolence. Then all critiques are assuaged.

**Resubmission Of Major Revision:**

The authors may consider submitting a major revision at a later time.